# Efficient-LVSM: Faster, Cheaper, and Better Large View Synthesis Model via Decoupled Co-Refinement Attention

**Xiaosong Jia**[1,2*], **Yihang Sun**[3*], **Junqi You**[3*], **Songbur Wong**[3], **Zichen Zou**[1,2],
**Junchi Yan**[3†], **Zuxuan Wu**[1,2†], **Yu-Gang Jiang**[1,2]
[1]Institute of Trustworthy Embodied AI (TEAI), Fudan University
[2]Shanghai Key Laboratory of Multimodal Embodied AI
[3]Sch. of Computer Science & Sch. of Artificial Intelligence, Shanghai Jiao Tong University
* Equal Contributions       † Correspondence Authors
https://efficient-lvsm.github.io/

## Abstract

Feedforward models for novel view synthesis (NVS) have recently advanced by transformer-based methods like LVSM, using attention among all input and target views. In this work, we argue that its full self-attention design is suboptimal, suffering from quadratic complexity with respect to the number of input views and rigid parameter sharing among heterogeneous tokens. We propose **Efficient-LVSM**, a dual-stream architecture that avoids these issues with a decoupled co-refinement mechanism. It applies intra-view self-attention for input views and self-then-cross attention for target views, eliminating unnecessary computation. Efficient-LVSM achieves 29.86 dB PSNR on RealEstate10K with 2 input views, surpassing LVSM by 0.2 dB, with 2× faster training convergence and 4.4× faster inference speed. Efficient-LVSM achieves state-of-the-art performance on multiple benchmarks, exhibits strong zero-shot generalization to unseen view counts, and enables incremental inference with KV-cache, thanks to its decoupled designs.

## 1 Introduction

Reconstructing 3D scenes from a collection of 2D images remains a cornerstone challenge in computer vision. The field has witnessed a remarkable evolution, moving from classical photogrammetry systems to per-scene optimized neural representations like NeRF (Mildenhall et al., 2020) and 3DGS (Kerbl et al., 2023), which achieve high-quality reconstruction, but require dense inputs and costly optimization for each new scene. A significant advance came from Large Reconstruction Models (LRMs) (Hong et al., 2024; Wei et al., 2024; Zhang et al., 2024), which learn generalizable 3D priors from vast datasets. A recent paradigm shift, pioneered by models like LVSM (Jin et al., 2025), has further advanced the field by minimizing hand-crafted inductive biases, where it directly synthesizes novel views from posed images. It eliminates the need for predefined 3D structures or rendering equations and achieves surprisingly good rendering quality with flexibility.

Despite the success, its monolithic self-attention mechanism, where all input and target tokens are concatenated into a single sequence, leads to two primary drawbacks: (1) Low efficiency: full self-attention leads to quadratic complexity with regard to the number of input views (Jia et al., 2023d). Furthermore, when generating multiple target views with the same input views, input representation can not be re-used. (2) Limited performance: full self-attention enforces parameter sharing for heterogeneous tokens - content-rich input views and pose-only target queries. It hinders the model's ability to learn specialized representations for their distinct tasks, i.e., understanding the semantics & 3D structure of the scene for input tokens and rendering the novel view for target tokens.

This work was supported by the Science and Technology Commission of Shanghai Municipality (No. 24511103100) and the New Cornerstone Science Foundation through the XPLORER PRIZE. This work was supported by Ant Group through CCF-Ant Research Fund. This work was also in part supported by Scientific Research Innovation Capability Support Project for Young Faculty (U40) of the Ministry of Education of China, SRICSPYF-ZY2025019.

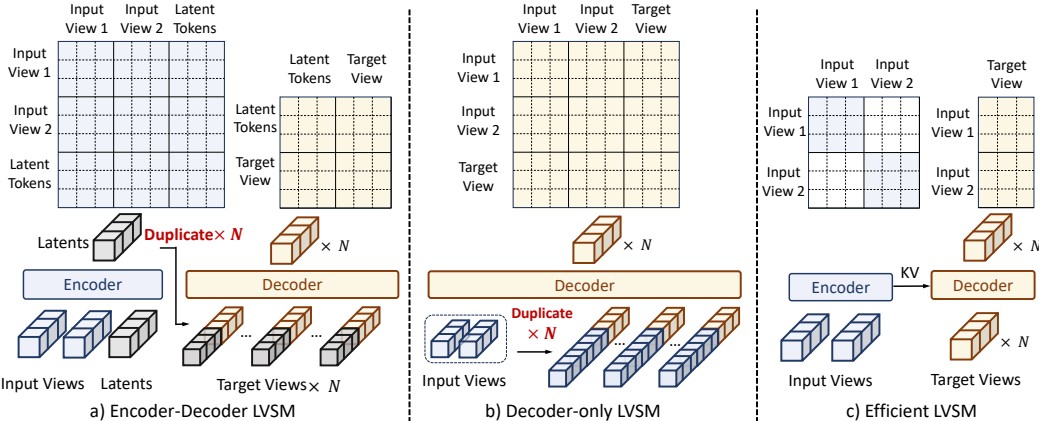

Figure 1: **Latent Novel View Synthesis Paradigms Comparison.** The proposed decoupled architecture disentangles the input and target streams with lower $O(N_{in})$ complexity and no duplication of tokens.

In this work, we systematically analyze these trade-offs and derive **Efficient-LVSM**, a Transformer-based architecture designed to resolve these limitations. The key insight is to **decouple** the process of input view encoding from target view generation (Jia et al., 2023b; Wu et al., 2022). To realize this, Efficient-LVSM employs a dual-stream architecture. First, a dedicated **Input Encoder** is solely responsible for processing the source views. It uses intra-view self-attention to independently build a representation of each input's content and geometry. Second, a **Target Decoder** is tasked with synthesizing the novel view. The Target Decoder engages in a continuous dialogue with the encoder: **at each layer**, the target tokens first refine their own understanding of the target view's structure through self-attention, and then query the features from the corresponding layer of the Input Encoder via cross-attention.

- **Specialized Attention Pathways.** Our architecture utilizes distinct modules for input and target tokens (Jia et al., 2023c). In the input encoder, only input view is processed. In the target decoder, target tokens act as queries and input tokens serve as keys and values in cross-attention, avoiding the use of shared parameters for heterogeneous information.

- **Robustness to Variable View Counts.** The input self-attention processes each view separately, making the transformation of one view independent of others. This per-view processing strategy allows the model to generalize better than LVSM to a variable number of input views at test time.

- **Computational and Memory Efficiency.** The input encoder processes each input view separately and the target decoder adopts cross-attention, both reducing the computational complexity with respect to the number of input views from **quadratic** $O(N_{in}^2)$ to **linear** $O(N_{in})$.

- **Incremental Inference via KV-Cache.** The decoupled structure enables KV-cache of input view features. When a new input view is provided, only that view needs to be processed. When a new target view is required, the KV-cache could be directly re-used. In summary, the cost of adding new input views and target views is nearly constant and thus enables incremental inference.

We conduct comprehensive evaluations for Efficient-LVSM. It sets a new state-of-the-art, outperforming LVSM by 0.2dB PSNR and GS-LRM by 1.7dB PSNR on the RealEstate10K benchmark with $50\%$ training time and achieves $2-4$ times speed acceleration in terms of both training iteration and inference. It exhibits strong zero-shot generalization to unseen numbers of input views.

## 2 METHOD

In this section, we present a step-by-step analysis that derives the design of Efficient-LVSM.

### 2.1 PRELIMINARY

**Task Definition:** Given $N$ input images with known camera poses and $M$ target view camera poses, novel view synthesis (NVS) aims to render $M$ corresponding target images. Specifically, the input is $\{(\mathbf{I}_i, \mathbf{E}_i, \mathbf{K}_i)|i = 1, 2, ..., N\}$ and $\{(\mathbf{E}_j, \mathbf{K}_j)|j = 1, 2, ..., M\}$, where $\mathbf{I} \in \mathbb{R}^{H \times W \times 3}$ is the input RGB image, $H$ and $W$ are the height and width, $\mathbf{E}, \mathbf{K} \in \mathbb{R}^{4 \times 4}$ are camera extrinsic and intrinsic. The output is rendered target images, denoted as $\{\hat{\mathbf{I}}_j|j = 1, 2, ..., M, \hat{\mathbf{I}} \in \mathbb{R}^{H \times W \times 3}\}$

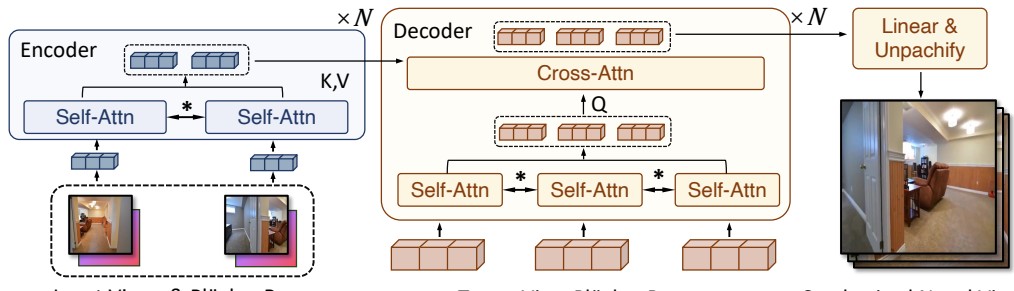

Figure 2: **Efficient-LVSM Model Structure.** Efficient-LVSM patchifies posed input images and target Plücker rays into tokens. Input tokens pass separately through an encoder to extract context, while target tokens cross-attend to generate new views. Asterisks indicate shared parameters.

**Feedforward NVS Framework:** we adopt LVSM (Jin et al., 2025) end-to-end paradigm. For the input $\{(\mathbf{I}_i, \mathbf{E}_i, \mathbf{K}_i)\}_{i=1}^{N}$ and $\{(\mathbf{E}_i, \mathbf{K}_i)\}_{i=1}^{M}$, all camera poses are encoded using Plücker ray embedding (Plucker, 1865) while input images are patchified as in ViT (Dosovitskiy et al., 2020). We obtain the input tokens $\{\mathbf{S_i}\}_{i=1}^{N}$ by concatenating its RGB patches and Plücker ray patches in the hidden dimension and passing through an MLP. We obtain the target tokens $\{\mathbf{T_j}\}_{i=1}^{M}$ by feeding its Plücker ray patches into another MLP.

Next, input and target tokens pass through a set of transformer blocks to extract features, which is the key component of the framework: $\{\mathbf{R_j}\}_{j=1}^{M} = \Phi(\{\mathbf{S_i}\}_{i=1}^{N}, \{\mathbf{T_j}\}_{j=1}^{M})$ where $\Phi$ represents the transformer blocks and $\{\mathbf{R_j}\}_{j=1}^{M}$ is the final features of target views.

The output layer transforms the final features of target views $\{\mathbf{R_j}\}_{j=1}^{M}$ into RGB value by a linear layer followed by a sigmoid function. The resulting RGB patches are then unpatchified and assembled to form the corresponding target images $\hat{\mathbf{I}}_j^T$, producing the final synthesized outputs:

$$\hat{\mathbf{I}}_j^T = \text{unpatchify}(\text{Sigmoid}(\text{Linear}_{\text{render}}(\mathbf{R}_j)) \in \mathbb{R}^{H \times W \times 3} \tag{1}$$

| Structure | Overall Complexity | Component | Complexity |
|---|---|---|---|
| LVSM Encoder-Decoder | $O(N^2 + M)$ | Encoder Decoder | $O(N^2)$ $O(M)$ |
| LVSM Decoder-Only | $O(M(N+1)^2)$ | Decoder | $O(M(N+1)^2)$ |
| Efficient-LVSM (**Ours**) | $O(NM + N)$ | Encoder Decoder | $O(N)$ $O(NM)$ |

Table 1: **Comparison of Model Structure Complexity.** The proposed Efficient-LVSM obtains lower complexity than LVSM and thus achieves significant speed up, as evidenced in Sec. 3.4.

## 2.2 ANALYSIS OF LVSM'S FULL SELF-ATTENTION PARADIGM

LVSM deocder-only model employs full self-attention on all input and target tokens, which introduces the following two limitations:

**Entangled Representation.** From the content perspective, input tokens contain both semantic and geometric information, while target tokens only have geometric information. From the system perspective, they bear distinct tasks: input tokens are to understand the semantics & 3D structure of the scene (Jia et al., 2023a) and target tokens are to render the novel view. However, shared self-attention parameters do not distinguish the difference (Yang et al., 2025a), hampering the generalization ability, as evidenced experiments in Table 2 and visualizations in Fig. 7c.

**Computation and Memory Costs.** Consider a sample $(\mathbf{S_i}, \mathbf{T_j})$ with the shapes of $NP \times d$ and $MP \times d$, where $N$ and $M$ are the numbers of input and target views, and $P = HW/p^2$ is the number of patches ($p$ represents the patch size). LVSM decoder-only model constructs $M$ separate sequences for $M$ target views, where each sequence is a concatenation of the entire set of input tokens and the tokens of a single target view. These sequences are processed by full self-attention across multiple transformer layers. For a given layer indexed by $l$, the operation is defined as:

$$\mathbf{V_i}^l = \text{concat}(\mathbf{S_1}^l, \mathbf{S_2}^l, ..., \mathbf{S}_N^l, \mathbf{T_j}^l); \quad \mathbf{V_i}^l = \mathbf{V_i}^{l-1} + \text{Self-Attn}_{\text{full}}^l(\mathbf{V_i}^{l-1}) \tag{2}$$

The shape of $\mathbf{V_i}$ is $M \times (NP + P) \times d$. LVSM repeats the computation of one target view for $M$ times. Thus, the temporal complexity of LVSM decoder-only model is $M \cdot O(N^2 P^2) = O(N^2 M)$,

as shown in Fig. 1 and Table 1. The quadratic complexity with regard to the number of input views hampers the efficiency and the repetition of tokens introduces severe computational cost.

LVSM encoder-decoder structure avoids the repetition issue by using an encoder to compress all input views into one latent vector first. However, this design introduces **loss of information**, significantly limiting the reconstruction quality, which is acknowledged in LVSM paper Jin et al. (2025).

## 2.3 DUAL-STREAM PARADIGM

Based on the observation above, we propose a dual-stream structure, where distinct modules are applied on input and target tokens to decouple the information flow, as in Fig. 2.

**Input Encoder:** To maintain the independency of different input views and improve efficiency, we limit the scope of self-attention to patches within the same input view. Each input view is processed separately, which enables efficient inference when a new input view is provided (incremental inference). Instead of constructing a single, prohibitively long attention sequence containing tokens from all N input views, we propose to process N shorter sequences. Specifically, let $\mathbf{S_i}$ represent the tokens of the $i^{th}$ input view. They are updated by an intra-view self-attention block at layer $l$:

$$\mathbf{S_i}^l = \mathbf{S_i}^{l-1} + \text{Self-Attn}_{\text{input}}^l(\mathbf{S_i}^{l-1}); \quad \mathbf{S_i}^l = \mathbf{S_i}^l + \text{FFN}_{\text{input}}^l(\mathbf{S_i}^l) \tag{3}$$

**Target Decoder:** To allow efficient KV-cache for features of input views, target decoder employs cross-attention, letting output tokens $\mathbf{T_j}^l$ attend to input tokens $\mathbf{S_i}^L$ from the last layer of encoder:

$$\mathbf{T_j}^l = \mathbf{T_j}^l + \text{Cross-Attn}_{\text{target}}^l(\mathbf{T_j}^l, \mathbf{S_1}^L, \mathbf{S_2}^L, ..., \mathbf{S_N}^L); \quad \mathbf{T_j}^l = \mathbf{T_j}^l + \text{FFN}_{\text{input}}^l(\mathbf{T_j}^l) \tag{4}$$

This design decouples the parameters for input and output tokens with dual-stream structure and allows rendering multiple target views with the same input KV-cache. While this approach shares conceptual similarities with recent architectures like MM-DiT (Esser et al., 2024), which use different projections for heterogeneous inputs, a key distinction lies in our architectural choice. We argue that our tokens differ not just in content but in their fundamental computational roles: inputs are content-rich providers, while targets are content-agnostic queries. This asymmetry motivates our use of a fully decoupled dual-stream architecture, in contrast to MM-DiT's unified, full self-attention block.

Assuming the hidden dimension and the number of patches per image are constants, the temporal complexity of the Target Decoder are $O(NM)$, while complexity of LVSM decoder-only is $O(M(N+1)^2)$, as in Table 1.

## 2.4 INTRA-VIEW ATTENTION OF TARGET TOKENS IN DECODER

The aforementioned cross-attention only decoder design introduces a drawback: **each target token has to store the information of the whole scene by their own** since there is no scene-level interaction in input encoder, limiting the capacity. To this end, we propose to add intra-view self-attention in target decoder alternatively with the original cross attention :

$$
\begin{aligned}
\mathbf{T_j}^l &= \mathbf{T_j}^{l-1} + \text{Self-Attn}_{\text{target}}^l(\mathbf{T_j}^{l-1}) \\
\mathbf{T_j}^l &= \mathbf{T_j}^l + \text{Cross-Attn}_{\text{target}}^l(\mathbf{T_j}^l, \mathbf{S_1}^l, \mathbf{S_2}^l, ..., \mathbf{S_N}^l) \\
\mathbf{T_j}^l &= \mathbf{T_j}^l + \text{FFN}_{\text{input}}^l(\mathbf{T_j}^l)
\end{aligned}
\tag{5}
$$

In this way, the intra-view self-attention in decoder allows to integrate scene-level information from other target tokens while still maintaining KV-cache ability. Experiments Table 6 (a) demonstrates 6+6 layers self-then-cross attention performs better than 12 layers cross-attention.

## 2.5 CO-REFINEMENT OF ENCODER-DECODER

One widely observed phenomenon for deep neural network is that **different layers of features represent different abstract level of informantion** (Zeiler & Fergus, 2013): early layers capturing fine-grained details such as textures, and later layers encoding high-level semantics. In vanilla encoder-decoder, only last layer features are used, as in Fig. 3 (a), which limits its capacity.

To this end, we propose a dual-stream co-refinement structure, illustrated in Fig. 3 (b), where each layer of the encoder provides information to its corresponding layer in the decoder. At layer $l$, input tokens $\mathbf{S}^l$ are first updated by self-attention, and then the target decoder queries these updated tokens

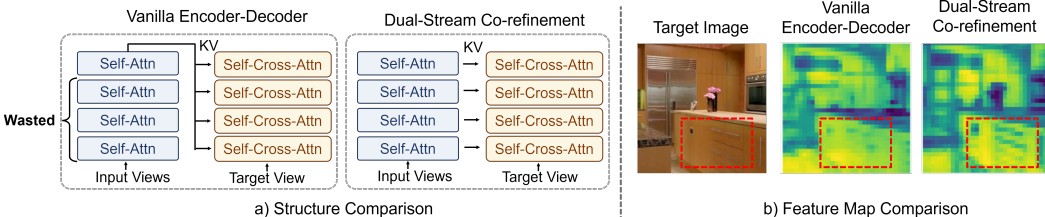

Figure 3: **Vanilla Encoder-Decoder vs. Dual-Stream Co-refinement.** (a) Hidden features in middle layers in vanilla encoder-decoder are wasted while the dual-stream co-refinement structure utilizes these features to extract more information. (b) Feature maps indicate that co-refinement structure catches more details of the target view.

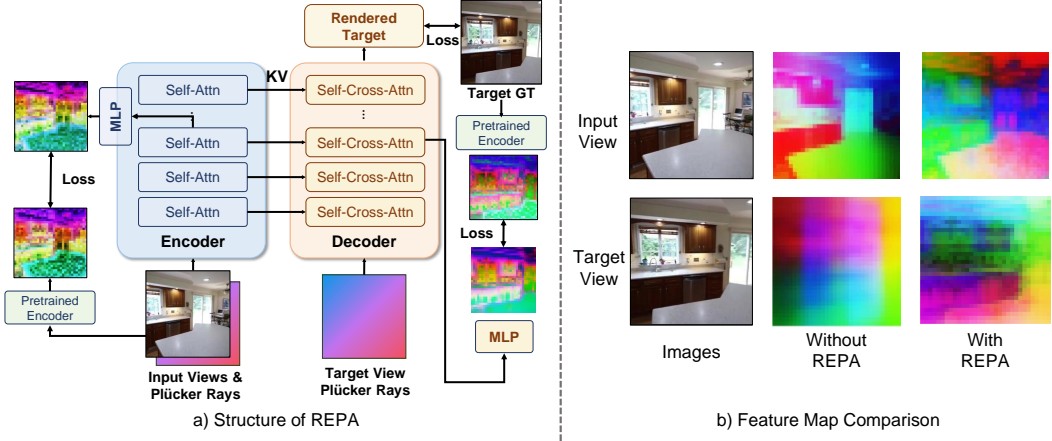

Figure 4: **Applying REPA into Efficient-LVSM.** (a) Pretrained vision encoders and MLP projectors are discarded in inference. (b) Feature maps indicate that REPA helps the model extract semantics.

to refine its own representation $\mathbf{T}^l$:

$$
\begin{aligned}
\mathbf{T_j}^l &= \mathbf{T_j}^{l-1} + \text{Self-Attn}^l_{\text{target}}(\mathbf{T_j}^{l-1}) \\
\mathbf{T_j}^l &= \mathbf{T_j}^l + \text{Cross-Attn}^l_{\text{target}}(\mathbf{T_j}^l, \mathbf{S_1}^l, \mathbf{S_2}^l, ..., \mathbf{S}_N^l) \\
\mathbf{T_j}^l &= \mathbf{T_j}^l + \text{FFN}^l_{\text{input}}(\mathbf{T_j}^l)
\end{aligned}
\tag{6}
$$

By querying the encoder's representations in the middle layers, the decoder can synthesize its own features using both the fine-grained details from early layers and the rich semantic context from later ones. Fig. 3 (b) demonstrates that the co-refinement model generate more detailed and high-quality features compared to vanilla encoder-decoder structure.

## 2.6 DISTILLATION WITH REPA

With the decoupled attention for different views, a natural thought is to utilize those powerful pretrained vision encoder. To utilize visual features without sacrificing inference speed, we employ REPA (Yu et al., 2025) to distill visual features from DINOv3 (Siméoni et al., 2025). Formally, consider a clean image $\mathbf{I}$ and $h_\phi(\mathbf{X_k})$ is the projection of hidden features of layer $k$, where $h_\phi$ is a trainable projector and $\mathbf{X_k}$ represents the input tokens or target tokens of layer $k$: $\mathbf{X_k} = \mathbf{S_k}$ or $\mathbf{X_k} = \mathbf{T_k}$. Let $f$ represent the pretrained encoder such as DINOv3. The goal is to align the projection of layer output $h_\phi(\mathbf{X_k})$ with encoded images $f(\mathbf{I})$ by maximizing the patch-wise similarities:

$$
\mathcal{L}_{REPA} = \frac{1}{N} \sum_{i=1}^{N} \text{sim}(f(\mathbf{I}), h_\phi(\mathbf{X_k}))
\tag{7}
$$

We find that improvement with REPA is conditional. Experiment in Table 6 show that LVSM benefits much less compared to Efficient LVSM's dual-stream co-refinement design structure, possibly due to its full self-attention design entangles feature maps of different views.

## 2.7 KV-CACHE & INCREMENTAL INFERENCE

A key advantage of the decoupled dual stream design is its natural compatibility with KV caching during inference as illustrated in Fig. 10. The key and values of all input views, $\{\hat{\mathbf{S}}_i\}_{i=1}^N$, can be

Table 2: **Scene-level View Synthesis Quality.** We test on the same validation set proposed in pixelSplat.

| | RealEstate10k (Zhou et al., 2018) | | |
|---|---|---|---|
| | PSNR ↑ | SSIM ↑ | LPIPS ↓ |
| pixelNeRF | 20.43 | 0.589 | 0.550 |
| ViewCrafter | 21.63 | 0.642 | 0.175 |
| GPNR | 24.11 | 0.793 | 0.255 |
| SEVA | 25.66 | 0.841 | 0.139 |
| Du et al. | 24.78 | 0.820 | 0.213 |
| pixelSplat | 26.09 | 0.863 | 0.136 |
| DepthSplat | 27.46 | 0.889 | 0.115 |
| MVSplat | 26.39 | 0.869 | 0.128 |
| GS-LRM | 28.10 | 0.892 | 0.114 |
| LVSM Enc-Dec(res-256) | 28.58 | 0.893 | 0.114 |
| LVSM Dec-Only(res-256) | 29.67 | 0.906 | 0.098 |
| **Ours(res-256)** | 28.93 | 0.895 | 0.102 |
| LVSM Enc-Dec(res-512) | 28.55 | 0.894 | 0.173 |
| LVSM Dec-Only(res-512) | 29.53 | 0.904 | 0.141 |
| **Ours(res-512)** | 29.86 | 0.905 | 0.147 |

Table 3: **Object-level View Synthesis Quality.** We test at 512 and 256 resolution on both input and rendering. "Enc" means encoder and "Dec" means decoder.

| | ABO (Collins et al., 2022) | | | GSO (Downs et al., 2022) | | |
|---|---|---|---|---|---|---|
| | PSNR ↑ | SSIM ↑ | LPIPS ↓ | PSNR ↑ | SSIM ↑ | LPIPS ↓ |
| Triplane-LRM (Res-512) | 27.50 | 0.896 | 0.093 | 26.54 | 0.893 | 0.064 |
| GS-LRM (Res-512) | 29.09 | 0.925 | 0.085 | 30.52 | 0.952 | 0.050 |
| LVSM Enc-Dec (Res-512) | 29.86 | 0.913 | 0.065 | 29.32 | 0.933 | 0.052 |
| LVSM Dec-Only (Res-512) | 32.10 | 0.938 | 0.045 | 32.36 | 0.962 | 0.028 |
| **Ours (Res-512)** | 32.65 | 0.951 | 0.042 | 32.92 | 0.973 | 0.021 |
| LGM (Res-256) | 20.79 | 0.813 | 0.158 | 21.44 | 0.832 | 0.122 |
| GS-LRM (Res-256) | 28.98 | 0.926 | 0.074 | 29.59 | 0.944 | 0.051 |
| LVSM Enc-Dec (Res-256) | 30.35 | 0.923 | 0.052 | 29.19 | 0.932 | 0.046 |
| LVSM Dec-Only (Res-256) | 32.47 | 0.944 | 0.037 | 31.71 | 0.957 | 0.027 |
| **Ours (Res-256)** | 33.13 | 0.960 | 0.035 | 32.73 | 0.969 | 0.022 |

computed once and stored. When a new target view is required, the decoder could directly utilize the stored cache $\{\hat{\mathbf{S}}_i\}_{i=1}^{N}$ for rendering. When a new input view $\mathbf{I}_{N+1}$ is introduced, only this new view needs to be processed and appended into the cache. As a result, it enables efficient incremental inference, which could be used in interactive application scenarios.

## 3 EXPERIMENTS

### 3.1 DATASETS

**Scene-level Datasets.** We use the widely used RealEstate10K dataset (Zhou et al., 2018). It contains 80K video clips curated from 10K YouTube videos, including both indoor and outdoor scenes. We follow the training/testing split applied in LVSM (Jin et al., 2025).

**Object-level Dataset.** We use the Objaverse dataset (Deitke et al., 2023) to train our model. Following the rendering settings in GS-LRM (Zhang et al., 2024), we render 730K objects, and each object contains 32 random views. We test our object-level model on Google Scanned Objects (Downs et al., 2022) (GSO) and Amazon Berkeley Objects (Collins et al., 2022) (ABO), containing 1099 and 1000 objects respectively. Following Instant-3D (Li et al., 2023b) and LVSM (Jin et al., 2025), we render 4 structured input views and 10 random target views for testing.

### 3.2 IMPLEMENTAION DETAILS

**Model Details.** Following LVSM (Jin et al., 2025), we use a patch size of $8 \times 8$ for the image tokenizer with 24 transformer layers (12-layer encoder and 12-layer decoder) and the dimension of hidden feature 1024. Following REPA (Yu et al., 2025), we select a 3-layer MLP as the alignment projection layer.

**Protocols.** Following the settings in LVSM, we select 4 input views and 8 target views in the object-level dataset. We select 2 input views and 3 target views in scene-level dataset.

More implementation details are provided in Appendix C.

### 3.3 COMPARISON WITH START-OF-THE-ART MODELS

**Scene-Level Comparison.** We compare on scene-level inputs with pixelNeRF (Yu et al., 2021), ViewCrafter (Yu et al., 2024), GPNR (Suhail et al., 2022), SEVA (Zhou et al., 2025a), Du et al. (Du et al., 2023), pixelSplat (Charatan et al., 2024), DepthSplat (Xu et al., 2025), MVSplat (Chen et al., 2024), GS-LRM (Zhang et al., 2024), LVSM encoder-decoderand LVSM decoder-only (Jin et al., 2025). As in Table 2, our model establishes a new state-of-the-art on the RealEstate10K benchmark, outperforming the previous leading method, LVSM decoder-only, by a significant margin of 0.2 dB PSNR. This corresponds to an $4.5\%$ reduction in Mean Squared Error (MSE), indicating a substantial improvement in reconstruction fidelity. This quantitative leap is supported by our qualitative results in Figure 5, where our model produces noticeably sharper renderings and demonstrates superior geometric accuracy, particularly when synthesizing near-field objects where LVSM often introduces artifacts. Notably, this state-of-the-art performance is achieved with remarkable efficiency. Our model was trained for just 3 days on 64 A100 GPUs, which is **half the training time**

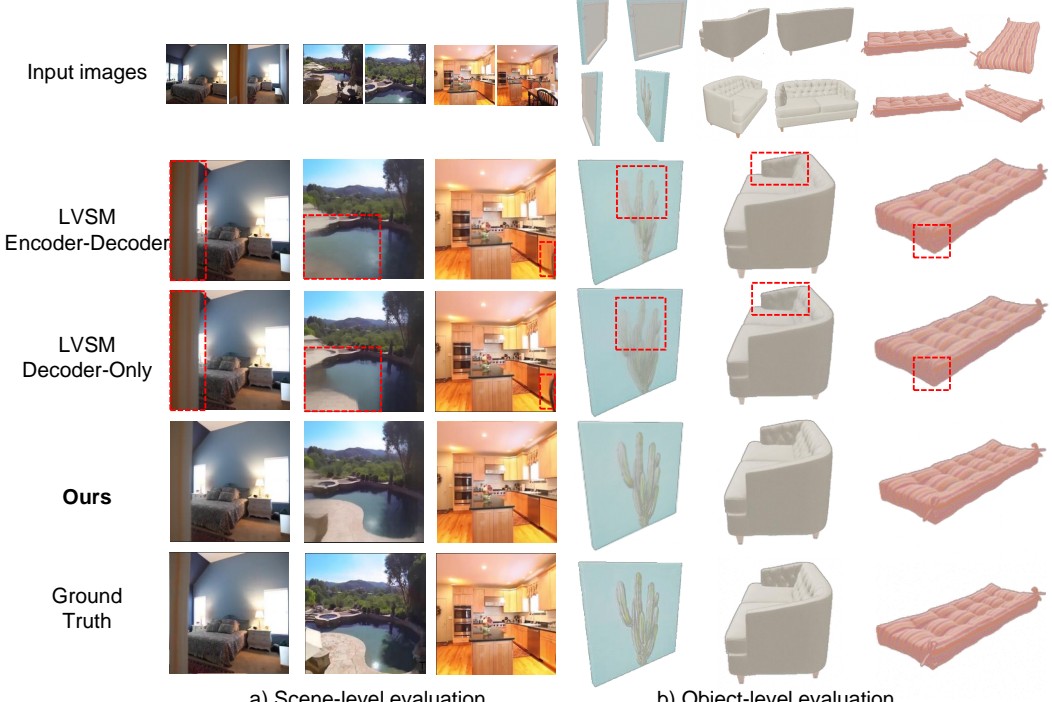

a) Scene-level evaluation    b) Object-level evaluation

Figure 5: **NVS Visual Comparison.** We compare with LVSM (Jin et al., 2025) in RealEstate10K (Zhou et al., 2018) and Amazon Berkeley Objects (Collins et al., 2022). Images rendered by our model have less blur details.

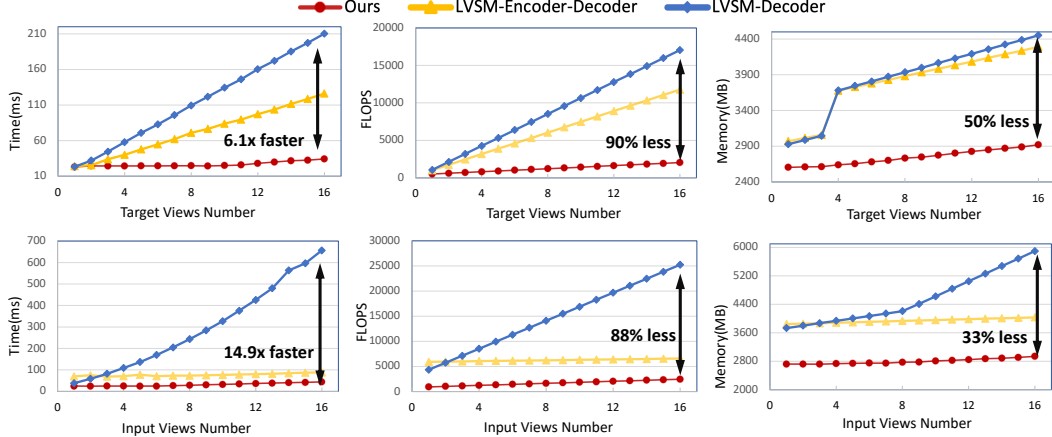

Figure 6: **Inference Speed Comparison.** We compare the inference time (ms) against (a) the number of target views and (b) the number of input views. Our model achieves consistently low latency. The performance of the LVSM baselines, particularly LVSM Decoder-Only, degrades severely as view counts increase. This highlights our model's significant computational efficiency, achieving up to a 14.9x speedup over LVSM Decoder-Only.

**required by LVSM**. In essence, Efficient-LVSM not only surpasses the previous state-of-the-art in quality but does so while **requiring only** $50\%$ **of the training budget**.

**Object-Level Comparison.** Similarly, Efficient-LVSM achieves state-of-the-art performance.

### 3.4 EFFICIENCY ANALYSIS

We evaluate the efficiency from three perspectives: vanilla inference latency, incremental inference latency, and training convergence speed. For fair comparison, we keep the number of layers (12+12) and hidden dimension (1024) the same with LVSM. For the convergence analysis, smaller variants are used for fast verification to save computational resource.

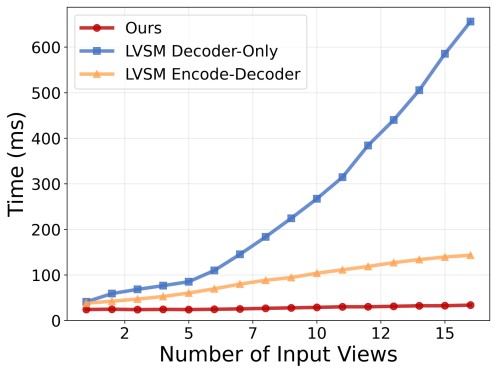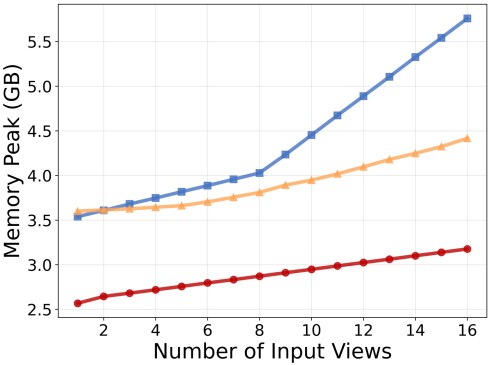

(a) **Incremental Inference Experiments.** We compare the inference latency and memory consumption when the input view is fed one by one. We observe that Efficient LVSM achieves near constant latency and memory consumption due to its KV-cache ability.

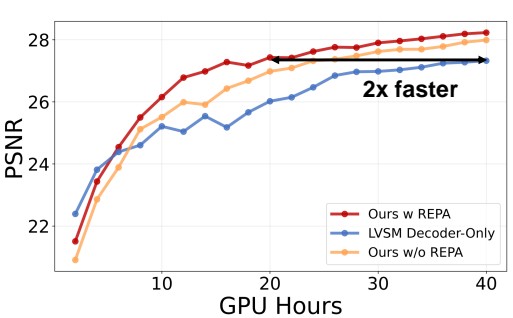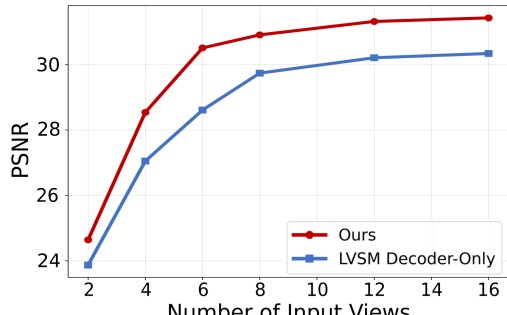

(b) **Training Speed Comparison.** Efficient LVSM delivers roughly 2× faster training and consistently achieves higher PSNR.

(c) **Zero-Shot Generalization to Input View Count.** Trained with 4 input views and tested on varying numbers of input views.

Figure 7: **Overall Comparison.**

Table 4: **Comparison of Different Novel View Synthesis Methods.**

| Type | Model | Param. | Latency (ms) ↓ | GFLOPS ↓ | PSNR ↑ |
|---|---|---|---|---|---|
| Optimization-Based | pixelNeRF | 28M | 2500+ | - | 20.43 |
| | GPNR | 10M | 6000+ | - | 24.11 |
| Feed-forward GS | pixelSplat | 125M | 50.52 | 1934 | 26.09 |
| | MVSplat | 12M | 10.23 | 583 | 26.39 |
| | GS-LRM | 307M | 88.24 | 5047 | 28.10 |
| Diffusion-based | SEVA | 2333M | 29000 | - | 27.46 |
| | ViewCrafter | 2609M | 38000 | - | 21.63 |
| Feed-forward Latent | LVSM Enc-Dec | 177M | 70.88 | 6042 | 28.58 |
| | LVSM Dec-Only | 177M | 109.37 | 8523 | 29.67 |
| | Ours (inference) | 199M | 24.78 | 1325 | 29.82 |

**Vanilla Inference Speed.** We analyze the inference cost by measuring latency, memory peak, and total GFLOPS as a function of both input and target view counts. As shown in Fig. 6, our model's resource consumption exhibits a slow growth, maintaining high efficiency even with many views. In contrast, while the LVSM Encoder-Decoder shows a moderate increase in cost, the LVSM Decoder-Only variant suffers from a severe computational growth. This efficiency gap is substantial and becomes increasingly pronounced as the number of views grows due to our linear complexity design. Specifically, with 16 input views, our model is approximately 14.9x faster and consumes 50% less memory than LVSM Decoder-Only. This demonstrates that our decoupled attention mechanism effectively removes the computational bottleneck caused by quadratic complexity.

**Incremental Inference.** Fig. 7a indicate that the time and memory required for the incremental coming input views is nearly constant for Efficient-LVSM. Conversely, both LVSM baselines exhibit a clear growth in latency and memory consumption.

Table 5: **Ablation Study of REPA Distillation.**

| Category | Configuration | PSNR ↑ | LPIPS ↓ | SSIM ↑ |
|---|---|---|---|---|
| **Without REPA Distillation (Baseline)** | | 26.02 | 0.1481 | 0.8483 |
| *Ablation on REPA Hyperparameters* | | | | |
| Loss Function | Smooth L1 | 26.81 | 0.1349 | 0.8562 |
| | L2 | 26.39 | 0.1366 | 0.8571 |
| | Cosine | 26.30 | 0.1374 | 0.8542 |
| Distillation Target | Input Tokens Only | 26.35 | 0.1367 | 0.8569 |
| | Target Tokens Only | 26.27 | 0.1452 | 0.8536 |
| | Both Input & Target | 26.60 | 0.1256 | 0.8642 |
| DINOv3 Source Layer | Layer 8 | 26.60 | 0.1256 | 0.8642 |
| | Layer 10 | 26.28 | 0.1441 | 0.8540 |
| | Layer 12 | 26.11 | 0.1416 | 0.8503 |

Table 6: **Ablation Study.**

(a) **Architectural Components Ablation.**

| Arch. | PSNR ↑ | SSIM ↑ | LPIPS ↓ |
|---|---|---|---|
| Cross-Attention Only | 24.18 | 0.7908 | 0.1982 |
| Self-Cross Attention | 24.97 | 0.8201 | 0.1628 |
| Co-Refinement | 26.25 | 0.8462 | 0.1490 |

(b) **Effect of REPA**

| Arch./Variant | PSNR ↑ | SSIM ↑ | LPIPS ↓ |
|---|---|---|---|
| LVSM Dec-Only | 25.52 | 0.8385 | 0.1541 |
| LVSM Dec-Only w REPA | 25.68 | 0.8410 | 0.1515 |
| Ours w/o REPA | 26.02 | 0.8483 | 0.1481 |
| Ours w REPA | 26.81 | 0.8628 | 0.1296 |

(c) **Effect of Model Sizes**

| Models | Parameters | PSNR ↑ | SSIM ↑ | LPIPS ↓ | Latency(ms)↓ | GFLOPS↓ | Memory ↓ |
|---|---|---|---|---|---|---|---|
| Enc(12) + Dec(12) | 199M | 28.32 | 0.8892 | 0.1106 | 24.78 | 1325 | 3032 |
| Enc(6) + Dec(6) | 101M | 27.77 | 0.8871 | 0.1149 | 17.58 | 647 | 1802 |
| Enc(3) + Dec(3) | 53M | 26.43 | 0.8609 | 0.1377 | 10.16 | 310 | 1321 |

**Training Speed.** As in Fig. 7 (b), Efficient-LVSM demonstrates a steeper learning curve. It successfully reaches the final performance plateau of the LVSM baseline while consuming only **half the computational budget (GPU hours)**.

**Comparison with State-of-the-art Paradigms**. To further assess the efficiency of Efficient LVSM within the broader landscape of novel view synthesis, we compare it against representative methods across four paradigms, as summarized in Table 4. As shown in the table, optimization-based methods (e.g., pixelNeRF (Yu et al., 2021)) require per-object optimization before rendering, resulting in substantial computation time. Diffusion-based models (e.g., ViewCrafter (Yu et al., 2024)) suffer from prohibitive latency (ranging from seconds to minutes) due to the large number of sampling steps, making them unsuitable for real-time applications. Although feed-forward Gaussian Splatting approaches such as MVSplat (Chen et al., 2024) provide extremely low latency, they typically sacrifice reconstruction quality.

Notably, our model surpasses GS-LRM (Zhang et al., 2024) in both quality and efficiency, requiring only ~26% of its GFLOPS. Compared to LVSM variants, Efficient-LVSM maintains the high-quality rendering characteristic of large latent models and reduce inference latency to 24.78 ms. This demonstrates that our decoupled attention design successfully mitigates the computational bottleneck of monolithic transformers without sacrificing performance.

### 3.5 ZERO-SHOT GENERALIZATION TO THE NUMBER OF INPUT VIEWS.

As in Fig. 7c, Efficient-LVSM and LVSM both could benefit from more views even not trained under such data, thanks to the set operator - Transformer. Efficient LVSM constantly outperforms LVSM under all view settings while the gap is gradually reduced, since the reconstruction becomes easier with more input views.

### 3.6 ABLATION STUDIES

All ablation experiments use a smaller 6+6 encoder-decoder configuration to save budget.

**Co-refinement of Encoder-Decoder Structure.** As in Table 6 (a), self-then-cross attention yields 0.79 dB PSNR improvement compared to cross-attention only in decoder. Further, adopting encoder-decoder co-refinement gives 1.28 dB PSNR gains.

**Applicability of REPA Distillation.** As in Table 6 (b), applying REPA to Efficient-LVSM brings a substantial gain of 0.8 dB while applying to LSVM only brings 0.16 dB improvement. In Table 5, we study the configuration of REPA. We find that Smooth L1 loss works the best, possibly due to its absolute approximation to DINOv3 features instead of relative approximation as cosine similarity. We confirm that distillation for both input and target are useful. DINOv3's middle layer features instead of the final layers are most helpful, aligning with findings in Siméoni et al. (2025).

**Influcne of Model Size.** As in Table 6 (c), increasing model size consistently improves reconstruction quality, aligning with Jin et al. (2025), demonstrating the potential of feedforward models.

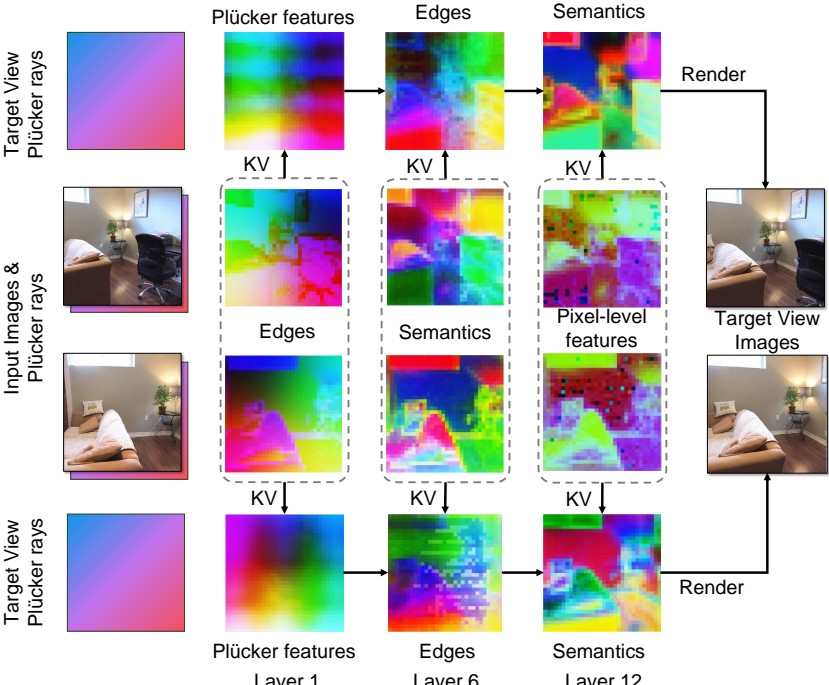

Figure 8: **PCA Visualization of Input and Target Views Features at Different Layers.**

## 4 VISUALIZATION

In Fig. 8, we visualize the features of Efficient-LVSM trained on RealEstate10K. We could observe that from the initial layer (Layer 1) to the middle layer (Layer 6), the features contain more and more semantics. From the middle layer (Layer 6) to the last layer (Layer 12), the features becomes similar to the final output - RGB images. The evolving process demonstrates the effectiveness of the proposed co-refinement structure to extract features from all levels. By bridging the encoder and decoder at each layer, the model ensures that fine-grained structural details are preserved while progressively extracting information of the scene.

## 5 CONCLUSION

In this work, we present a systematic analysis for issues of existing Transformer based NVS feedforward model, identifying the bottlenecks of monolithic attention, Based on the analysis, we derive Efficient LVSM, a decoupled dual-stream architecture that integrates an encoder-decoder co-refinement mechanism. Comprehensive experiments demonstrate that the proposed structure not only performs better but also achieves significant speed up for training convergence and inference latency. By successfully reconciling the trade-off between quality and cost, our approach paves the way for more scalable and interactive 3D generation applications.

**Ethics Statement.** Our research aims to advance the field of computer vision and does not present immediate, direct negative social impacts. We believe our work has the potential for a positive impact by improving. The dataset used in this study is publicly available and have been widely adopted by the community for academic research. All data was handled in accordance with their specified licenses and terms of use. We did not use any personally identifiable or sensitive private information. We have focused our evaluation on standard academic benchmarks. We encourage future research building upon our work to consider the specific ethical implications of their target applications.

**Reproducibility Statement.** To ensure the reproducibility of our research, we provide a comprehensive description of our methodology, implementation details, and experimental setup in the paper. Furthermore, we commit to making our code, pre-trained models, and experiment configurations publicly available upon publication of this paper.

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

## A    USE OF LARGE LANGUAGE MODELS (LLMS) STATEMENT

During the preparation of this manuscript, we utilized Large Language Models (LLMs), as a writing assistance tool. The use of LLMs was limited to improving the grammar, clarity, and readability of the text. This includes tasks such as rephrasing sentences for better flow, correcting spelling and grammatical errors, and ensuring stylistic consistency. The core scientific ideas, experimental design, results, and conclusions presented in this paper are entirely our own. LLMs were not used to generate any of the primary scientific content or interpre- tations. The final version of the manuscript was thoroughly reviewed and edited by all authors, who take full responsibility for its content and originality.

## B    RELATED WORKS

**Generalizable Novel View Synthesis.** The ability to synthesize novel views from a sparse set of images is a long-standing goal in computer vision. Pioneering approaches such as image-based rendering (IBR) blend reference images based on proxy geometries (Debevec et al., 1996; Gortler et al., 1996). Early deep learning based methods predict blending weights or depth maps (Hedman et al., 2018; Choi et al., 2019). Generalizable neural radiance fields models like PixelNeRF (Yu et al., 2021) and MVSNeRF (Chen et al., 2024) pioneered the use of 3D-specific inductive biases.

**Transformer-based Large Reconstruction Models.** The Transformer architecture (Vaswani et al., 2017), originally proposed for natural language processing, has demonstrated remarkable versatility and scalability across diverse domains (Jia et al., 2025b;a; Yang et al., 2025a; 2023; Fan et al., 2025). Transformer representations have been studied for geometry-aware perception and spatial reasoning under sparse multi-sensor settings Li et al. (2023a); Zhu et al. (2025); Jia et al. (2025c), suggesting that explicit handling of heterogeneous geometric cues is often beneficial.Building on its success, recent research has focused on developing Transformer-based Large Reconstruction Models (LRMs)(Hong et al., 2024; Wei et al., 2024; Li et al., 2023b; Gao et al., 2024; You et al., 2025) to capture robust and generalizable 3D priors from extensive datasets. These models are trained on vast datasets to learn generic 3D priors. For instance, Triplane-LRM(Li et al., 2023b) and GS-LRM (Zhang et al., 2024) learn to map sparse input images to explicit 3D representations like triplane NeRFs or 3D Gaussian Splatting primitives. Recent studies show that the choice of tokenization scheme can significantly affect the efficiency–quality trade-off in sequence modeling and generation Jia et al. (2024a).

**View Synthesis without Explicit 3D Representations.** A compelling line of research explores the possibility of performing novel view synthesis in a purely "geometry-free" manner. Early attempts such as Scene Representation Transformers (SRT) (Sajjadi et al., 2022), introduced the idea of using a Transformer to learn a latent scene representation. Large View Synthesis Model (LVSM) (Jin et al., 2025) employs a single, monolithic Transformer to process all input and target tokens.

**Generative and Diffusion-based Novel View Synthesis.** Parallel to deterministic approaches, generative models, particularly those based on diffusion (Watson et al., 2022; Zhou et al., 2025a), have gained traction for their ability to plausibly complete unobserved regions. A key challenge has been ensuring multi-view consistency. Solutions include test-time distillation into 3D representations like NeRFs (Poole et al., 2022; Wu et al., 2023b), or building consistency directly into the model using video backbones and cross-view attention (Gao* et al., 2024; Zhou et al., 2025a). This research

area also includes lightweight methods like NVS-Adapter (amd Jinwoo Lee et al., 2024), which adapts pre-trained 2D models for the single-view NVS task. Related efforts also explore autoregressive scene generation in other 3D sensing modalities, highlighting the broader interest in scalable generative scene modeling Zhou et al. (2025b).

## C  TRAINING AND IMPLEMENTATION DETAILS

**Training Setup.** We train Efficient-LVSM with a constant learning rate of 4e-4 with a warmup of 2500 iterations. Following LVSM (Jin et al., 2025), we use AdamW optimizer and the $\beta_1$ and $\beta_2$ are 0.9 and 0.95 respectively. We also employ a weight decay of 0.05 on all parameters of the LayerNorm layers. Unless noted, our models have 12 encoder layers and 12 decoder layers, which is the same as LVSM.

**Dataset-Specific Schedules.** For the object-level dataset, we use 4 input views and 8 target views, training on 64 A100 80G GPUs. We first train at a resolution of 256, using a batch size of 12 for 3 days, totaling 250k iterations. We then finetune the model at a resolution of 512 for 2 days, with a batch size of 2 per GPU and 100k iterations, using a learning rate of $4 \times 10^{-5}$. For the scene-level dataset, we use 2 input views and 3 target views. Training begins at a resolution of 256 with a batch size of 16 per GPU for 2 days, completing 650k iterations. We then finetune the model at a resolution of 512 for 1 day, using a batch size of 4 and 200k iterations. In the ablation studies, we train and evaluate on the RealEstate10K dataset using 2 input views and 3 target views. These models are trained on 2 A100 80G GPUs for 10 hours, with the per-GPU batch size set as large as permitted by memory capacity. For the model-size ablations, training is extended to 24 hours on 2 GPUs.

**REPA Distillation Details.** We use the DINOv3-ViT-B/16 model (Siméoni et al., 2025) as the pre-trained teacher. We use the output features from the 8th transformer layer of DINOv3 as the distillation target. These teacher features are aligned with the output of a specific layer in our student model, which varies by its size: for our main 12+12 layer models, we align with the 3rd layer's output, while for the smaller 6+6 layer models used in ablations, we align with the 2nd layer. The alignment is performed via a 3-layer MLP projector and optimized using the Smooth L1 loss (Girshick, 2015).

## D  ANALYSIS OF ALTERNATIVE ARCHITECTURAL DESIGNS

In this section, we evaluate our dual-stream co-refinement architecture by comparing it against several alternative designs. The idea of decoupling heterogeneous roles (e.g., context providers versus query executors) has been explored in prior work Hu et al. (2023); Lu et al. (2024); Wu et al. (2023a). Notably, the datapath of our architecture can, in principle, be reproduced using customized attention masks, and the separation of token roles (input vs. target tokens) may be approximated by assigning distinct projection layers within the attention module. To provide a comprehensive analysis, we implemented and assessed such alternatives.The alternative architectures are described below:

**LVSM w/ Mask**: We add a custom attention mask on a single Transformer backbone to emulate the information flow of our model: input tokens are restricted to self-attend only within themselves, while target tokens are permitted both self-attention and cross-attention over all input tokens.

**LVSM w/ MMDiT-style**: Inspired by multi-modal architectures (Esser et al., 2024), this variant uses a shared Transformer block but introduces separate projection layers ($W_q, W_k, W_v$) and feed-forward networks (FFNs) for input and target tokens, allowing the model to handle their distinct characteristics.

**LVSM w/ Mask + MMDiT-style**: A hybrid design that combines the custom attention mask with the specialized per-token-type projection layers within a single Transformer block.

We trained all variants on the RealEstate10K dataset while keeping the parameter budget as comparable as possible. Quantitative results are summarized in Table 7.

The results clearly highlight the advantages of our dual-stream co-refinement architecture. Although the LVSM w/ Mask variant uses fewer parameters, it is more than $7\times$ slower. This slowdown arises because irregular attention masks disrupt the highly optimized contiguous memory access patterns expected by underlying implementations; realizing any theoretical speedup would require custom CUDA kernels, undermining the simplicity and generality of the approach. Similarly, MMDiT-style

Table 7: **Comparison of Different Architectures.**

| Arch. | Parameters | PSNR ↑ | SSIM ↑ | LPIPS ↓ | Latency (ms) ↓ | GFLOPS ↓ | Memory ↓ |
|---|---|---|---|---|---|---|---|
| Corefinement | 101M | 26.02 | 0.8483 | 0.1481 | 17.58 | 647 | 1802 |
| LVSM | 86M | 25.24 | 0.8286 | 0.1545 | 74.45 | 3487 | 3114 |
| LVSM w/ mask | 86M | 24.13 | 0.7925 | 0.1793 | 125.13 | 1050 | 5758 |
| LVSM w/ MMDiT | 164M | 24.37 | 0.8021 | 0.1607 | 78.58 | 3487 | 3857 |
| LVSM w/ mask+MMDiT | 164M | 23.24 | 0.7601 | 0.1913 | 130.76 | 1050 | 6544 |

Table 8: **Latency and Speedup with KV-Cache.**

| Number of Input Views | Ours w/ KV-Cache (ms) ↓ | LVSM Dec-Only (ms) ↓ | Speedup Factor ↑ |
|---|---|---|---|
| 4 | 24.37 | 123.1 | **5.1x** |
| 8 | 28.62 | 286.0 | **10.0x** |
| 16 | 42.96 | 801.7 | **18.7x** |
| 32 | 72.84 | 2592 | **35.6x** |
| 48 | 103.26 | 5408 | **52.4x** |
| 64 | 138.43 | 9231 | **66.7x** |

variants incur significantly higher parameter counts and computational costs due to the duplicated projection and FFN layers for each token type.

This ablation study demonstrates that our dual-stream co-refinement architecture achieves an effective balance of reconstruction quality, inference speed, computational efficiency, and implementation simplicity for novel view synthesis.

## E    DETAILS ON KV-CACHING

Many interactive applications require reusing computation across iterative queries, motivating cache-friendly architectures and incremental inference pipelines that have been explored in other sequential decision and closed-loop settings Li et al. (2025); Yang et al. (2025b); Jia et al. (2024a;b); You et al. (2024). In contrast to repeatedly recomputing context, our decoupled design enables KV-caching for input features and supports near-constant-cost updates when adding new inputs or targets. As discussed in the main paper, our architecture's compatibility with KV-caching is key to its efficiency. Figure 10 provides a visual workflow of this incremental inference process and contrasts it with the LVSM baseline.

## F    DETAILED MODEL ARCHITECTURE

Fig. 9 displays the detailed model architecture and the data path.

## G    LIMITATIONS AND FUTURE WORK

Despite the progress, we recognize several limitations that offer promising avenues for future research. A primary limitation, which our work shares with other large-scale feed-forward models, is the path to industrial-scale deployment. Although Efficient-LVSM makes substantial strides in reducing computational costs, translating these large Transformer-based architectures into production-level applications with stringent latency and memory constraints remains a formidable challenge. In line with the perspective of the original LVSM paper, our model can be viewed as a powerful proof of concept that further validates the potential of geometry-free, Transformer-based novel view synthesis.

Bridging the gap between this academic proof of concept and widespread industrial adoption will likely require significant innovations beyond architectural design. Future work in this direction could explore techniques such as model compression, network quantization, and knowledge distillation to create smaller, faster versions of these models without substantial degradation in quality. We are hopeful that our contribution in improving the baseline efficiency of the feed-forward paradigm serves as a beneficial step on the path toward making these models practical for real-world use.

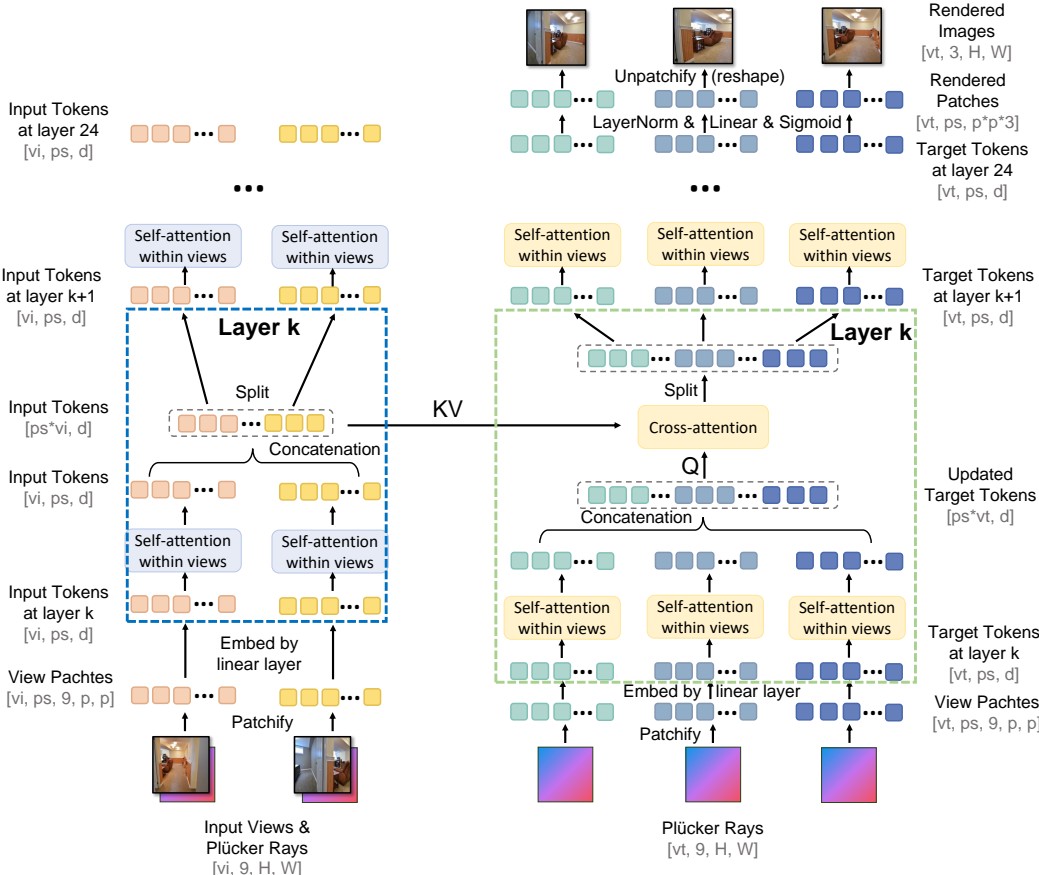

Figure 9: **Detailed Model Structure.** $ps = HW/p^2$ denotes the number of patches in an image, where $p$ is the patch size. $vi$ and $vt$ represent the numbers of input and target views, respectively. When $vi$ or $vt$ appears in the first (batch) dimension, it indicates that the tokens from different views are processed jointly as a batch.

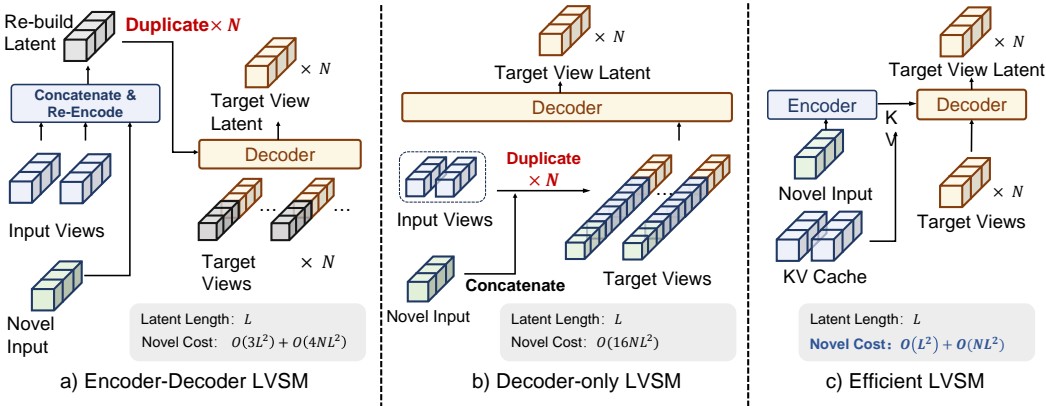

Figure 10: **Efficient Incremental Inference with KV-Cache.** Efficient LVSM saves computation when provided with novel inputs or targets by caching the key and value for previous input views.

