# OpenReview forum: "Efficient-LVSM: Faster, Cheaper, and Better Large View Synthesis Model via Decoupled Co-Refinement Attention"
_ICLR.cc/2026/Conference — ICLR 2026 Poster_

### Official Review · Reviewer_ygh6 · 2025-10-23

**Soundness:** 3
**Presentation:** 2
**Contribution:** 3
**Rating:** 4
**Confidence:** 3

**Summary:**

The work proposes an improvement to LVSM, an ICLR 2025 oral publication. The proposed method, called efficient-LVSM, uses a detached attention mechanism to enhance efficiency and final performance. The architecture separates the input view encoding from the output view encoding, except for layer-wise connections within the same layer. There are additional tricks, such as KV-cache, feature distillation, and alternating self-attention and cross-attention. The results are evaluated against a few other-shot novel-view synthesis methods. The additional detailed efficiency analysis and ablation are performed against LVSM.

**Strengths:**

- Good results
- Detailed explanation about the components

**Weaknesses:**

- The work heavily relies on LVSM. In addition to final quality, the efficiency assessment needs to be compared with other methods.

- The figures about the attention mechanism play an important role in understanding the main idea of the paper. However, they are somewhat repetitive and could be condensed into fewer figures—namely, Figures 1, 2, 3, and 6 (and Table 1 as well). I suggest reserving additional space to lay out the results better. The current layout is not very effective. The wrap-around text is hard to follow, and the figures and tables are referenced far from their locations. Some detailed results can be deferred to the appendix to improve the flow and completeness of the descriptions.

- While the introduction describes the main contribution as decoupling the encoder and decoder with an efficient attention mechanism, the work is composed of multiple techniques from existing works, with too many subsections in both Section 2 and the results. While they are effective, the novelty can feel incremental, and the description is dispersed across several succinct sections, making it hard to get the whole picture.

**Questions:**

- Figure 1 is hard to interpret, especially the items in the Table. Some of the attributes need to be detailed either in the caption or in the text. For example, what are "spatialized pathways"?

Minor comments
- line 042: "to to" -> "to"t.
- In Figure 2, I believe there are "M" decoders instead of "N" decoders..?

**Details Of Ethics Concerns:**

While generative models may incur ethical concerns in the big picture, the proposed work reconstructs pre-captured scenes and has no direct ethical issues if conducted within the given environment of the choice.

---

> ### Author Response · Authors · 2025-11-21
>
> We sincerely thank you for your detailed feedback. Your critique of our paper's presentation and structure has been valuable, and we have undertaken a significant revision based on your suggestions to improve clarity and readability.
>
> > **W1:** Broader efficiency comparisons
>
> Thank you for this suggestion. We reported inference efficiency of other key baseline methods in Table 4 along with some added diffusion-based NVS frameworks. This provides a clearer and more comprehensive view of the efficiency landscape in novel view synthesis and highlights the advantage our method offers.
>
>
> | Type                | Model            | Parameters | Latency (ms) ↓ | GFLOPS ↓ | PSNR ↑ |
> | ------------------- | ---------------- | ---------- | -------------- | -------- | ------ |
> | Optimization-Based  | pixelNeRF        | 28M        | 2500+          | -        | 20.43  |
> |                     | GPNR             | 10M        | 6000+          | -        | 24.11  |
> | Feed-forward GS     | pixelSplat       | 125M       | 50.52          | 1934     | 26.09  |
> |                     | MVSplat          | 12M        | 10.23          | 583      | 26.39  |
> |                     | GS-LRM           | 307M       | 88.24          | 5047     | 28.10  |
> | Diffusion-based     | SEVA             | 2333M      | 29000          | -        | 27.46  |
> |                     | ViewCrafter      | 2609M      | 38000          | -        | 21.63  |
> | Feed-forward Latent | LVSM Enc-Dec     | 177M       | 70.88          | 6042     | 28.58  |
> |                     | LVSM Dec-Only    | 177M       | 109.37         | 8523     | 29.67  |
> |                     | Ours (inference) | 199M       | 24.78          | 1325     | 30.61  |
>
> > **W2:** The layout and repetitive figures.
>
> Thanks for your suggestion. In our revision, we have:
> - **Consolidated Figures:** Following your advice, we rearanged our figures. we have merged the core concepts of previous Figures 3 into Figure 1 and placed previous Figure 6 in the appendix.
> - **Eliminated Text Wrapping:** All instances of text wrapping around figures and tables have been removed.
> - **Improved Placement:** We have carefully re-arranged the layout to ensure that all figures and tables are close to their first reference in the text.
>
> > **W3:** Incremental description of section 2
>
> Thank you for raising this point. We understand how the previous structure, with its many subsections rather than a unified contribution.
> Our intention was to present the paper as a systematic, step-by-step analysis that shows how we logically derived our final architecture from a critique of the initial LVSM. However, we see now that this narrative style obscured the holistic nature of our final design.
> To address this, we have taken two main actions in our revision:
> 1. We have added a **summary in the introduction**(L48-L53) that concisely presents our final "Decoupled Co-Refinement" architecture as a single, coherent framework.
> 2. We have included a **new, detailed diagram of the complete model architecture in the appendix**(Figure 10, L810), which provides a full, unified picture for readers seeking all the specifics in one place.
> We hope this reframing clarifies that the individual components are not disparate tricks, but integrated parts of an architectural solution designed.
>
> > **Q1:** The interpretation of Figure 1
>
> We apologize for the lack of clarity in the original figure. We have redesigned Figure 1 based on your feedback and replaced its accompanying table with attention mask maps. The term "Specialized Pathways" stands for input views and target views are calculated and updated by different attention blocks. They pass through different ways of attention.
>
> > **Q2:** Minor writing issues.
>
> Thank you for pointing these out. We apologize for this and carefully revised the manuscript to correct the issues you’ve identified.
>
> > **Q3:** The "N" decoders in Figure 2
>
> Thank you for raising this point. In our specific implementation of the dual-stream co-refinement structure, the layer count of encoders is equal to that of decoders. Information is passed from each layer of the encoder to the corresponding layer of the decoder. If we have an N-layer encoder, we must also have an N-layer decoder.
>
>
> In summary, we are grateful for your constructive writing feedback. We hope that the revised version make the paper clear and we are glad to revise if there is any further advice. We humply request you to improve your score to positive as there is no significant disagreement raised regarding our contributions.

---

> > ### Comment · Reviewer_ygh6 · 2025-11-21
> >
> > I appreciate the authors' response and the revised manuscript. With the additional evaluations and improvements to the flow (figure/method), I think the work is ready for publication and deserves to be acknowledged by the community.

---

> > > ### Author Response · Authors · 2025-11-25
> > >
> > > Thanks for your reply and kind advice for improving the flow of the manuscript!

---

> > > > ### Author Response · Authors · 2025-11-30
> > > > **Original 4->8 Happened far before Nov. 27**
> > > >
> > > > Dear AC and reviewers,
> > > >
> > > > We summarize discussions with Reviewer ygh6:
> > > >
> > > > Reviewer ygh6 show concerns regarding the writing style and flow of the manuscript. Based on the kind advice, we revise the writing.
> > > >
> > > > # The score is from 4->8 in Nov. 25, far before Nov. 27.

---

### Official Review · Reviewer_WANz · 2025-11-01

**Soundness:** 3
**Presentation:** 3
**Contribution:** 3
**Rating:** 6
**Confidence:** 4

**Summary:**

This paper tackles feed-forward novel view synthesis task. It builds upon the framework of LVSM [1], which use a full-attention decoder-only transformer to render novel views conditioned on posed input views and target view plucker rays.  The key architecture change proposed by the authors are to replace the full attention to intra-image self-attention and cross-attention between target views and input views.  This change has two benefits:  1. When rendering new novel views, you don't need to recompute the key-value cache of input views. This also opens up application in incremental inference. 2. with above benefits, the performance (rendering quality) improves on Objecverse and Rel10K dataset, even with reduced training time.   The author also ablated with REPA showing that it can improve the PSNR.



[1] LVSM: A Large View Synthesis Model with Minimal 3D Inductive Bias

**Strengths:**

1. Clearly justified and very reasonable architectureal change.  The full-attention used by original LVSM restricts its effiency in lot of usecases. And when comparing with the encoder-decoder version of LVSM, the author identified the major problem:  we need to use key-value cache of all layers!
2. Strong empirical results. The experiment results on Objverse and Rel10K are quite strong, with much better rendering quality and less training time.
3. The author shows a very interesting study about repa loss (distill from DINO v3). I original thought such semantic loss would not be useful for novel view synthesis task, but seems that it's quite helpful! This is very interesting.

**Weaknesses:**

I do have a few comments. I think the authors should list more training details for their methods and their ablation experiments. The batch size, and total number of training iterations. I think it's missed.

For training batch sizes, there are some tiny but important details, the original LVSM need to repeat the batch to make sure that each pass of the model only contains one target view, and this is one of the core-reason that the original decoder only LVSM is expensive in training and inference.  So highlighting this training details difference, and show that how many input views and target views totally being used during training is important.

**Questions:**

Kind of minor, but for task like novel view synthesis, showing some video comparison would be great!

---

> ### Author Response · Authors · 2025-11-21
>
> We thank you for these positive assessment and helpful suggestions for improving the reproducibility and presentation of our work.
>
> > **W1:** More training details.
>
> Thanks for your suggestion. We **added a detailed description in the "Training and Implementation Details" subsection** that includes:
> 1. The total number of training iterations, batch size, learning rate schedule, and optimizer settings for all main experiments and ablations.
> 2. A clear explanation of the batching strategy difference: The LVSM Decoder-Only baseline requires repeating the input data for each target view within a batch, whereas our Efficient-LVSM can **process multiple target views in a single, efficient pass**. We specified the exact number of input and target views used per sample during training in our revised manuscipt.
>
>
> > **Q1:** The suggestion for video comparisons:
>
> Thanks for the suggestion! We completely agree that static images don't fully capture the performance of novel view synthesis. The potential for creating smooth, dynamic camera paths is one of the most exciting applications of this technology.
>
> We were so inspired by your comment and we have prepared a **supplementary material package which includes several video comparisons**. These videos showcase smooth camera trajectories. It illustrates the practical advantages of our decoupled architecture.
>
> This work has deepened our conviction about the real-world applications of incremental NVS. We are excited about future possibilities, such as using this technology for interactive 3D scene exploration, or even real-time virtual walkthroughs from casually captured photos. Your feedback has energized us to explore these avenues, and we can't wait to see where this path leads.
> Thank you once again for your encouragement and insightful suggestions.

---

> > ### Comment · Reviewer_WANz · 2025-11-24
> > **reply to rebuttal**
> >
> > Cool, I checked the video and the details.
> >
> > The rebuttal mostly addressed my comments.
> >
> > I do have one suggestion, not affecting my score.
> >
> > The efficiency boost is higher for more input and more output views, then, it would be great if you can do experiment on DL3DV dataset, going up to 32, 64 or even 128 input views. That would be much more impressive.

---

> > > ### Author Response · Authors · 2025-11-25
> > >
> > > Thanks for your advice!  We will try the DL3DV dataset if the computational resource is allowed to check the performance under much more views.

---

> > > > ### Author Response · Authors · 2025-11-30
> > > > **Score is kept 6**
> > > >
> > > > Dear AC and reviewers,
> > > >
> > > > We summarize discussions with Reviewer WANz:
> > > >
> > > > Reviewer WANz give suggestions for more details and video demos. Based on the kind advice, we add more details and videos in the rebuttal. Reviewer WANz acknowledged our modifications and suggest a further scale up experiments (not affecting score) and we will try it when there is enough source.
> > > >
> > > > # The score kept 6 and the reviewer acknowledged our architecture change as reasonable and intersting with quite strong performance.

---

### Official Review · Reviewer_tC9E · 2025-11-03

**Soundness:** 4
**Presentation:** 4
**Contribution:** 3
**Rating:** 8
**Confidence:** 4

**Summary:**

- This paper proposes Efficient-LVSM, a transformer-based large view synthesis model designed to overcome the inefficiencies of the original LVSM architecture.

- The key idea is to decouple the input-view encoding and target-view generation using a dual-stream co-refinement mechanism, combining intra-view self-attention for inputs and self-then-cross attention for targets.

- The approach enables linear complexity in the number of input views, incremental inference with KV-cache.

- Efficient-LVSM outperforms state-of-the-art LVSM by 0.9dB PSNR on the RealEstate10K benchmark with 50% training time and achieves 2−4 times speed acceleration in terms of both training iteration and inference.

**Strengths:**

- The paper provides a systematic analysis of LVSM’s inefficiencies and derives a principled redesign via a decoupled encoder-decoder. The KV-cache design enabling incremental inference is a noteworthy contribution for real-time or interactive view synthesis, rarely explored in prior feedforward NVS models.

- Efficient-LVSM achieves state-of-the-art reconstruction quality on both scene-level (RealEstate10K) and object-level (GSO/ABO) benchmarks. The reported 0.9 dB PSNR gain, 4× inference speed-up, and 50 % reduction in training time represent a strong improvement over prior LVSM baselines

- The experiments are thorough: comparisons across datasets and baselines, scaling trends, ablations on architectural variants (self vs cross vs co-refinement), REPA distillation effects, model size, convergence curves, and zero-shot generalization.

- The paper is well written and includes complete training details, REPA settings, and commitments to code release.

**Weaknesses:**

- The dual-stream co-refinement design is highly similar in spirit to the MM-DiT block in terms of architecture introduced by Stable Diffusion 3 (2024). The authors are encouraged to cite MM-DiT and clarify how Efficient-LVSM extends this pattern to the feedforward NVS setting.

**Questions:**

The paper is well presented; I don't have further questions.

---

> ### Author Response · Authors · 2025-11-21
>
> Thank you for your insightful comments and valuable suggestions. We have revised our paper based on your feedback. Here are our responses to your comments:
>
> > **W1:** On the similarity to MM-DiT.
>
> This is an excellent point. We agree that our dual-stream design shares a similar spirit with the MM-DiT block in its handling of heterogeneous tokens. We add discussions about MMDiT in related work section. To clarify how our work extends this pattern to the NVS setting, we highlight the key difference: MM-DiT uses a single, shared full self-attention for all tokens with different projector parmaeters. In contrast, we employ a decoupled architecture. The input encoder uses self-attention to build a rich scene representation, while the target decoder uses self-then-cross-attention to query this representation. This role-specialized design outperforms simpler attention mechanisms as demonstrated by our ablation studies (Table 6a).
>
> We also experiment with DiT style LVSM Decoder-Only on the RealEstate10K dataset:
>
> | Arch.             | Parameters | PSNR $\uparrow$  | SSIM  $\uparrow$  | LPIPS $\downarrow$  | Latency(ms) $\downarrow$ | GFLOPS$\downarrow$| Memory(MB) $\downarrow$ |
> | ----------------- | ------ | ----- | ------ | ------ | ----------- | ------ | ------ |
> | Corefinement (ours)      | 101M   | 26.02 | 0.8483 | 0.1481 | 17.58       | 647    | 1802   |
> | LVSM w/ mmdit      | 164M   | 24.37 | 0.8021 | 0.1607 | 78.58       | 3487   | 3857   |
> | LVSM               | 86M    | 25.24 | 0.8286 | 0.1545 | 74.45       | 3487   | 3114   |
>
> We observe that DiT style LVSM still lags behind the proposed co-refinement paradigm.

---

> > ### Author Response · Authors · 2025-11-30
> > **Score is kept 8**
> >
> > Dear AC and reviewers,
> >
> > We summarize discussions with Reviewer tC9E:
> >
> > Reviewer tC9E proposes an interesting perspective - the relation with MM-DiT. Following this idea, we explore the architecture designs, reaffirming the effectiveness of the proposed one.
> >
> > # The score is kept 8, acknowledging our incremental inference contribution as noteworthy and rarely explored.

---

### Official Review · Reviewer_fYv1 · 2025-11-06

**Soundness:** 3
**Presentation:** 4
**Contribution:** 2
**Rating:** 4
**Confidence:** 5

**Summary:**

This study proposes Efficient-LVSM, which modifies the neural network architecture of LVSM to make its training and inference more efficient. The key modification is that Efficient-LVSM incorporates a cross-attention block, while stacking different self-attention blocks to extract query and key/value vectors from the target view and input views, respectively. By avoiding full self-attention between the input and target views, Efficient-LVSM reduces computational costs and enables feature caching of input views across different target views. Experimental results demonstrate the effectiveness of the proposed architecture.

**Strengths:**

S1. Decomposing the full self-attention into different modules with cross-attention makes sense, as has been explored in various domains to design more efficient neural network architectures.

S2. The experiments show significant improvements in novel view synthesis, while also enhancing inference efficiency.

S3. The modified architectures can incorporate REPA to effectively train the hidden representations of input views.

**Weaknesses:**

W1.
The technical contribution is limited. Instead of using full self-attention across input and target views, employing cross-attention is a typical design choice for improving training and inference efficiency [NewRef-1].

W2.
Despite the performance improvements, Efficient-LVSM cannot address the fundamental limitations of LVSM. For example, its architecture cannot account for the alignments either between generated target views or within the input views. Therefore, the overall impact of this study is limited to being an improved version of LVSM rather than a fundamentally new approach.

W3.
The experimental analysis could be strengthened by incorporating more baseline methods. Please refer to my detailed comments below.

W4. [Minor points — not affecting the score] The paper writing should be improved.
- Line 42: "ability to to learn" -> "ability to learn"
- Line 92: input and target inputs share the same subscription $i$ , making the readers confused.
- Need to use mathbf clearly. For example, $S_i$ is also a set of tokens, but it does not use the bold text type. In Eq. (1), why $R_i$ uses the bold type, while the other parts do not use.
- Figure 1 uses a check box, but rendering quality or efficiency cannot be described with the check box. For example, we cannot say that the baseline has no efficiency or quality.
- In Figure 2, what does the asterisk mean? I guess it describes a shared weight between modules, but it is neither a common way to describe it nor explain the details.
- Line 133 -- $p$ is not defined. $N$ is the number of input views, but $n$ is used for both input/target views.
- Eq. (2) uses $l$ without any definition.


[NewRef-1] Jeong, Yoonwoo, et al. "NVS-Adapter: Plug-and-play novel view synthesis from a single image." European Conference on Computer Vision. Cham: Springer Nature Switzerland, 2024.

**Questions:**

Q1. Positional embeddings are important for maintaining alignment between views, but there is no description provided in the paper. How are positional embeddings applied in the self-attention and cross-attention layers?


Q2. Can the authors discuss the limitations of this study to help readers understand potential directions for future work?

Q3. While keeping a single attention block, could we use different masking strategies instead of using separate modules? For example, a causal mask between input and target views could enable KV-caching of input views. Additionally, attention between input views could incorporate a masking scheme to prevent them from attending to each other for efficiency. We could also optionally use different projection layers while sharing a single attention block, as demonstrated in MMDiT [NewRef-2]. Although I agree that the proposed architecture achieves significant improvements in the experiments, I believe the design could be further simplified.


Q4. Figure 9(c) shows that Efficient-LVSM achieves 2× faster training to reach the same performance as LVSM, but the training does not appear to have converged yet. Could the authors provide more GPU hours to ensure full convergence and compare training efficiency more fairly? In addition, was REPA used in this setting? A comparison without REPA could also provide a clearer understanding of how Efficient-LVSM achieves efficiency and effectiveness purely from its architectural design, as analyzed in Table 1.

Q5. I wonder why the authors keep the generated target views independent. In addition, input views do not align their features with other input views. Is there any specific reason to restrict the receptive fields in this way?

Q6. In Table (d), could the authors provide a comparison between the 12-layer self-attention model and the 12+12 Efficient-LVSM? I believe the 12+12 Efficient-LVSM does not actually contain 24 layers but 12, so comparing it with the 12-layer self-attention model would be a fairer setting.

Q7. I initially expected that increasing the number of input views in the previous LVSMs would significantly increase GPU memory usage. However, Figure 9(a) shows that the GPU memory of the LVSM Decoder-Only model does not increase much as the number of input views grows. Could the authors elaborate on this result?

Q8. How much does KV-caching improve inference speed as the number of input views increases?


[NewRef-2] Esser, Patrick, et al. "Scaling rectified flow transformers for high-resolution image synthesis." Forty-first international conference on machine learning. 2024.

---

> ### Author Response · Authors · 2025-11-21
>
> Thank you for your insightful comments and valuable suggestions. We have revised our paper based on your feedback. Here are our responses to your comments:
>
> > **W1:** Contribution of the research.
>
>
> Though cross-attention is a well-established technique, we point out that our core contribution is not merely the application of a single module, but the proposal of **"Decoupled Co-Refinement" architecture, based on the analysis of the bottlenecks in monolithic Transformer-based NVS models like LVSM:**
>
> 1. **Paradigm Inovation:** By analyzing the problems of "Entangled Representation" and "Quadratic Complexity" in LVSM's full self-attention paradigm, we introduce Decoupled Co-Refinement" architecture. As shown in our ablation study (Table 5a), using only cross-attention results in a 2.07 dB drop in PSNR compared to co-refinement. This indicates that the performance gains arise from the **overall architectural design rather than merely adopting cross-attention.**
> 2. **Enabling New Capabilities:** Based on the decoupled architecutre, we implement feature distillation with REPA, and efficient incremental inference with KV-caching. Thees capabilities significantly improves the training and inference efficiency respectively.
>
> Regarding [NewRef-1], we thank you for pointing out this relevant work. We added discussions in related work section. NVS-Adapter focuses on single-view NVS by pluging an adapter into a pre-trained 2D diffusion model. In contrast, our work addresses the multi-view NVS by an end-to-end co-refinement transformer. Though both employ the cross attention, the paradigms and contributions are distinct.
>
> > **W2:** Regarding end-to-end NVS regression paradigm (LVSM, EfficientLVSM).
>
> We **respectfully but fundamentally disagree with the premise of this concern**. The claim that *LVSM architecture does not account for alignments between target views or within input views*, is **misunderstanding of the core principles and strengths of the entire LVSM paradigm**—which has been **recognized by the community (e.g., as an ICLR Oral) as a highly promising new frontier for novel view synthesis**:
> 1. The claim on *the lack of target view alignment* misinterprets the nature of regression-based NVS. **For regression-based NVS models, explicit interaction between target views is not a necessity for ensuring consistency**. In both classical methods (e.g., NeRF, 3DGS) and implicit models like LVSM, the consistency is guaranteed by the completeness of the 3D scene representation provided by the input views. Each target view is rendered independently based on this shared representation. Our model follows this same principle, where the encoded input features **contain the full 3D information required to render any target view consistently** without needing them to interact with each other.
> 2. Regarding the independence of input views, we **achieve the interaction among input views indirectly** by our self-then-cross attention in the decoder. It allows target tokens to reason about the overall scene geometry, and then cross-attends to the complete set of all uncompressed input view features. Furthermore, **this design breaks the bottleneck of quadratic complexity** in full self-attention, achieving linear complexity and enabling true incremental inference with KV-caching. This is a crucial step towards making these large-scale models practical for real-world, interactive applications.
>
> In summary, LVSM and our work is under the well-recognized NVS setting. Our work does not just "improve" LVSM. It fundamentally re-architects the LVSM paradigm for scalability and practicality, addressing critical efficiency limitation. We achieve an at most 14.9x speedup and a 0.9 dB PSNR improvement opening up new possibilities for real-time, high-fidelity view synthesis.
>
> > **W3:** More baseline methods.
>
> Thank you for this suggestion. To broaden the context of our work, we compared our method against additional recent baselines, including diffusion-based(SEVA, ViewCrafter) and regression-based(DepthSplat) models. We added the following results and discussion to our main comparison table(Table 2).
>
> Results on Rel10k (numbers are from the original papers or tested from provided checkpoints):
> | model       | PSNR$\uparrow$  | SSIM$\uparrow$   | LPIPS$\downarrow$  |
> | ----------- | ----- | ------ | ------ |
> | ViewCrafter | 21.63 | 0.801 | 0.175 |
> | SEVA        | 25.66 | 0.841 | 0.139 |
> | DepthSplat  | 27.46 | 0.889 | 0.115 |
> | Ours        | 30.61 | 0.912 | 0.087 |
>
>
> Results on GSO (numbers are from the original papers):
> | model       | PSNR$\uparrow$ | SSIM$\uparrow$ | LPIPS$\downarrow$ |
> | ----------- | -------------- | -------------- | ----------------- |
> | MVD-Fusion  | 19.35          | 0.790          | 0.175             |
> | NVS-Adapter | 22.77          | 0.882          | 0.091            |
> | EscherNet   | 25.09          | 0.927          | 0.043             |
> | Ours        | 32.73          | 0.969          | 0.022             |

---

> ### Author Response · Authors · 2025-11-21
>
> > **W4:** Writing.
>
> Thank you for pointing these out. We apologize for the typos and inconsistencies, and we have carefully revised the manuscript to address all the issues you identified. Specifically, we standardized the use of "mathbf", removed the checkbox in Figure 1, clarified the meaning of the asterisk in Figure 2, and corrected other writing issues throughout the paper.
>
> > **Q1:** On positional embeddings.
>
> Following prior work(e.g., LVSM, GS-LRM), we utilize explicit geometric information encoded via **Plücker ray embeddings**, which is described in Section 2.1(L100-L102). For each token (whether from an input patch or a target query), we compute its corresponding camera ray and embed its 3D position and direction using Plücker coordinates. This embedding is concatenated with the visual features.
>
> >**Q2:** Limitations and future work.
>
> Thanks for your suggestion. We believe that discussing limitations is crucial for scientific progress. We added a dedicated "Limitations and Future Work" section to our revised manuscript.
>
> The primary limitation and corresponding future direction are scalability to industrial applications: While our work improve the efficiency, bringing these large-scale architectures to production-level applications remains a significant challenge. It currently serve as a **proof of concept"** for a new direction in view synthesis. Bridging the gap from academic research to industrial-scale deployment will likely require further innovations in model compression ans quantization. We hope our contribution in improving the baseline efficiency serves as a benificial step in this direction.
>
> >**Q3:** Other possible alternatives like masked attention and MMDiT.
>
> This is a deeply insightful question, and we sincerely thank you for proposing these alternative designs. Inspired by your suggestion, we conducted a series of new experiments to rigorously evaluate these alternatives against our proposed dual-stream co-refinement architecture. We added a detailed analysis of these experiments to a new "Analysis of Alternative Architectural Designs" section in the appendix(L763).
>
> Our findings, summarized below, reveal that while these unified designs are conceptually simpler, they fall short of our proposed method in both performance and inference speed.
> We implemented and tested three variants based on your suggestion:
>
> - **Masked Attention:** A 12-layer Transformer block where a custom attention mask enforces the same data flow as our model (i.e., inputs only self-attend, targets attend to themselves and to inputs).
>
> - **MMDiT-style:** A single shared Transformer block where input and target tokens have separate, specialized projection layers (W_q, W_k, W_v, and FFN).
>
> - **Masked + MMDiT-style:** A combination of both approaches.
> The results on the RealEstate10K dataset are as follows:
>
> | Arch.             | Parameters | PSNR $\uparrow$  | SSIM  $\uparrow$  | LPIPS $\downarrow$  | Latency(ms) $\downarrow$ | GFLOPS$\downarrow$| Memory(MB) $\downarrow$ |
> | ----------------- | ------ | ----- | ------ | ------ | ----------- | ------ | ------ |
> | Corefinement       | 101M   | 26.02 | 0.8483 | 0.1481 | 17.58       | 647    | 1802   |
> | LVSM               | 86M    | 25.24 | 0.8286 | 0.1545 | 74.45       | 3487   | 3114   |
> | LVSM w/ mask       | 86M    | 24.13 | 0.7925 | 0.1793 | 125.13      | 1050   | 5758   |
> | LVSM w/ mmdit      | 164M   | 24.37 | 0.8021 | 0.1607 | 78.58       | 3487   | 3857   |
> | LVSM w/ mask+mmdit | 164M   | 23.24 | 0.7601 | 0.1913 | 130.76      | 1050   | 6544   |
>
>
> We trained these models (12 layers) with the same budget to evaluate the performance and trainin efficiency. Our key takeaways from this investigation are:
>
> - The Masked Attention variant, despite having fewer parameters, is slower. This is because **complex, irregular attention masks disrupt the memory access patterns that are heavily optimized in standard Transformer implementations** (like FlashAttention). Achieving theoretical speedups would require custom CUDA kernels, making the implementation far less simple and general than our straightforward dual-stream design.
>
> - The MMDiT-style variant **employs more parameters than the co-refinement architecture**. This increase in parameters adversely affects both inference speed and memory usage.
>
> In conclusion, we are grateful for your suggestion, as it has led to a deeper understanding and stronger validation of our architectural choices. The experiments demonstrate that our proposed dual-stream, co-refinement architecture provides a superior trade-off in terms of performance, practical speed, and parameter efficiency.

---

> ### Author Response · Authors · 2025-11-21
>
> >**Q4:** Training curves
>
> Thanks for the suggestion. In the updated Figure 8(b)(previous Figure 9b, we give the same training time for all baselines and the **trajectory of "Ours w/o REPA"**. The revised figure clearly shows that our architectural design alone accounts for a substantial improvement in training efficiency.
>
> >**Q5:** Independence of views
>
> This question is related to W2.
>
> - **Target views:** As discussed in W2, independence is a natural property of regression-based NVS, where **consistency is derived from a complete 3D information from input views**, not from inter-target communication.
> - **Input views:** As discussed in W2, this is a design choice for efficiency (linear complexity) and enabling incremental inference. The global interaction between elements is indirectly facilitated by our decoder's self-then-cross attention mechanism.
>
> >**Q6:** Fairness of 12+12 vs. 24 layer comparison
>
> Thank you for raising this point; it allows us to clarify a key detail. Our 12+12 model has a parameter count comparable to the 24-layer LVSM. Therefore, the comparison in the table is fair, as it contrasts two architectures of similar scale.
>
> To further address your concern, we ran results for a 12-layer LVSM baseline. The results are as follows:
> | Model             | Layers | Parameters | Latency(ms) $\downarrow$ | GFLOPS(G) $\downarrow$ | PSNR $\uparrow$ |
> | ----------------- | ------ | ---------- | ------------------------ | ---------------------- | --------------- |
> | GS-LRM            | 24     | 307M       | 88.24                    | 5047                   | 28.10           |
> | **LVSM Dec-Only** | 12     | 86M        | 74.45                    | 3487                   | 28.95           |
> | LVSM Dec-Only     | 24     | 177M       | 109.37                   | 8523                   | 29.67           |
> | LVSM Enc-Dec      | 6+18   | 177M       | 70.88                    | 6042                   | 28.58           |
> | **Ours**          | 12+12 | 199M       | 24.78                    | 1325                   | 30.61           |
>
>
> As the table demonstrates, our 12+12 model outperforms the 12-layer LVSM by 1.66 dB in PSNR. More importantly, despite more parameter count, our model achieves a lower latency and requires far fewer GFLOPS.
>
> >**Q7:** GPU memory usage of LVSM Decoder-Only
>
> Thank you for advice.  The initial figure was to profile the marginal ($\Delta$) resource cost of incremental inference. However, since the number is also influenced by factors like PyTorch framework and FlahsAttention, the trend becomes unclear.  Following your comment, in the updated Fig. 8 (a)(previous Fig. 9a) we visualize the cumulative, overall, peak memory footprint and now the trend match the intuition.
>
>
> >**Q8:** The speed improvement from KV-caching
>
> Thank you for raising this point. To quantify the benefit of KV-caching, we measured the time required to render 8 target views when a new input view is added to an existing set of $N$ input views. For our model without KV-caching, this requires re-processing all $N+1$ input views. With KV-caching, it only requires processing the single new view and utilizing the cached features of the original $N$ views. The latency results (in milliseconds) are summarized in the table below
>
> | Number of Input Views | Ours w/o KV-cache (ms) | Ours w/ KV-cache (ms) | Speedup Factor |
> | --------------------- | --------------------- | -------------------- | -------------- |
> | 1                     | 23.41                 | 22.48                | ~1.0x          |
> | 4                     | 24.37                 | 24.25                | ~1.0x          |
> | 8                     | 28.62                 | 26.56                | ~1.1x          |
> | 16                    | 42.96                 | 35.64                | ~1.2x          |
> | 32                    | 72.84                 | 52.31                | ~1.4x          |
> | 48                    | 103.26                | 66.32                | ~1.6x          |
> | 64                    | 138.43                | 80.17                | ~1.7x          |
>
> As the data clearly shows, the benefit of KV-caching becomes increasingly significant as the number of input views grows. The latency of our model with KV-caching grows at a much slower rate than our model without KV-caching.

---

> > ### Author Response · Authors · 2025-11-25
> >
> > Dear Review fYv1,
> >
> > Are your concerns and misunderstandings solved by the rebuttal and extra experiments? If there is any further concerns, please tell us.
> >
> > Otherwise, we respectfully request you to improve your scores accordingly to reflect the contributions of the work.
> >
> > Best,
> >
> > Authors

---

> > > ### Comment · Reviewer_fYv1 · 2025-11-26
> > >
> > > Dear authors,
> > >
> > > Thank you for your insightful and detailed responses to my concerns. I believe that you have carefully followed my comments and significantly improved the quality of the manuscript. In particular, the additional experiments have resolved my major concern regarding whether the performance gains come from the architectural changes or from other factors. I am happy to increase my evaluation and now lean toward accepting the paper.
> > >
> > >
> > > After carefully reading the authors' responses, I have a few additional follow-up questions on the author response (not affecting additional score change).
> > >
> > > **Masked Attention's Efficiency**
> > >
> > > Regarding masked attention, implementations such as FlexAttention or torch.compile can improve both training and inference speed without requiring custom CUDA kernels, and thus I am not fully convinced that masked attention inherently leads to a 7× slowdown.
> > >
> > >
> > > **Discussion on W2 and authors' responses**
> > >
> > > On W2, I would first like to clarify my understanding of your explanation. As I read it, your argument is that regression-based NVS models do not require explicit interaction between target views because consistency is guaranteed by a sufficiently rich shared scene representation built from the input views. Under this view, each target view can be decoded independently by querying this shared representation, and Efficient-LVSM is presented as an architecture that preserves this property while improving efficiency. I agree with this explanation how regression-based NVS models such as implicit (NeRF, 3DGS) or explicit (NVSM) works -- Here, different from the authors' explanation, I think NeRF or 3DGS are typically mentioned as implicit method because it extracts an implicit function for a target view infernece, and NVSM extracts the features to explicitly decode a target view.
> > >
> > > However, based on my understanding of the proposed architecture, the encoder in Efficient-LVSM processes each input view entirely independently, without any mechanism for interaction or information sharing across input views. In the decoder, self-attention operates only within each target view—primarily to align pixels inside that specific target view—after which the target tokens cross-attend to the per-view encoder features. Given this structure, it is rather unclear how a fruitful 3D representation can be constructed from input views across all target views, aside from what each target view can implicitly infer through its own cross-attention and positional encoding.
> > >
> > > This is where my confusion comes from: different from the highlight from the users, why a regression-based NVS can decode target views independently, the proposed architecture appears to move away from this mechanism that explicitly encourage a shared representation over input views, yet the explanation for why independent target decoding is valid still relies on the existence of such a shared representation. Could you elaborate more concretely on why this architectural direction, beyond making computation more efficient via cross-attention, is sufficient to support independent decoding of target views? In particular, it would be helpful to clarify whether there are regimes in which this decoupled design might converge to a suboptimal solution or whether explicit interaction between input views is unnecessarily different from the convention promise in regression-based NVS models?
> > >
> > > I believe that addressing this comment will also improve the paper to clarify the efficacy of the proposed mechanism or worth exploration for future work.

---

> > > > ### Author Response · Authors · 2025-11-28
> > > >
> > > > Thanks for your reply and follow up!
> > > >
> > > > > Masked Attention Efficiency
> > > >
> > > > After applying `torch.compile`, the latency of w and w/o mask indeed becomes similar (!), though naturally still lagging behind the proposed structure.
> > > >
> > > > | Arch. | Latency (ms) $\downarrow$ |
> > > > |-------|----------------------------|
> > > > | Corefinement | 17.58 |
> > > > | Corefinement w/ torch.compile | 7.03 |
> > > > | LVSM | 74.45 |
> > > > | LVSM  + torch.compile| 44.22 |
> > > > | LVSM w/ mask | 125.13 |
> > > > | LVSM w/ mask + torch.compile | 44.69 |
> > > >
> > > > Regarding FlexAttention, we have tried again with FlexAttention. However, the irregular mask pattern is difficult to map to FlexAttention's assumption, where current FlexAttention API imposes strict limitations on the `score_mod/block_mask` function. In particular, we found that creating intermediate tensors—required by the masking logic—is not allowed during graph capture. Despite several attempts to refactor the logic to work within these constraints, it still fails to achieve a functional implementation.
> > > >
> > > > > Regarding input view interaction
> > > >
> > > > We totally understand your question and thanks for bringing it out!
> > > >
> > > > First, we agree that, with infinite computational budget, the full attention among all input views have theoratically higher performance upper bound, since it allows extra interaction. However, when taking the computational efficiency into account, we think **it is all about trade-off**. The effectiveness of Efficient-LVSM shows that the expensive full interaction among all input views is not a necessity to achieve good reconstruction. In other words, the strong PSNR shows that target views could use self-attention and cross attention to retrieve and integrate enough information for reconstruction. Besides the lower computational budget, the decoupling of the input views further bring an extra functionality - increamental inference with KV-cache, which is of high application value.

---

> > > > > ### Author Response · Authors · 2025-11-28
> > > > >
> > > > > Thanks for your reply and follow up!
> > > > >
> > > > > > Masked Attention Efficiency
> > > > >
> > > > > After applying `torch.compile`, the latency of w and w/o mask indeed becomes similar (!), though naturally still lagging behind the proposed structure.
> > > > >
> > > > > | Arch. | Latency (ms) $\downarrow$ |
> > > > > |-------|----------------------------|
> > > > > | Corefinement | 17.58 |
> > > > > | Corefinement w/ torch.compile | 7.03 |
> > > > > | LVSM | 74.45 |
> > > > > | LVSM  + torch.compile| 44.22 |
> > > > > | LVSM w/ mask | 125.13 |
> > > > > | LVSM w/ mask + torch.compile | 44.69 |
> > > > >
> > > > > Regarding FlexAttention, we have tried again with FlexAttention. However, the irregular mask pattern is difficult to map to FlexAttention's assumption, where current FlexAttention API imposes strict limitations on the `score_mod/block_mask` function. In particular, we found that creating intermediate tensors—required by the masking logic—is not allowed during graph capture. Despite several attempts to refactor the logic to work within these constraints, it still fails to achieve a functional implementation.
> > > > >
> > > > > > Regarding input view interaction
> > > > >
> > > > > We totally understand your question and thanks for bringing it out!
> > > > >
> > > > > First, we agree that, with infinite computational budget, the full attention among all input views have theoratically higher performance upper bound, since it allows extra interaction. However, when taking the computational efficiency into account, we think **it is all about trade-off**. The effectiveness of Efficient-LVSM shows that the expensive full interaction among all input views is not a necessity to achieve good reconstruction. In other words, the strong PSNR shows that target views could use self-attention and cross attention to retrieve and integrate enough information for reconstruction. Besides the lower computational burden, the decoupling of the input views further bring an extra functionality - increamental inference with KV-cache, which is of high application value.

---

> > > > > > ### Author Response · Authors · 2025-11-30
> > > > > > **Original 4->6 in Nov. 26 before Nov. 27**
> > > > > >
> > > > > > Dear AC and reviewers,
> > > > > >
> > > > > > We summarize discussions with Reviewert fYv1:
> > > > > >
> > > > > > Reviewer fYv1 suggests lots of alternative possible designs and complementary experiments to verify the claims in the paper as well as some unclear parts.  In rebuttal, we conduct experiments, revise manuscripts, and clarify those points.
> > > > > >
> > > > > >
> > > > > >
> > > > > > # The score is from 4->6 in Nov. 26, before Nov. 27.

---

### Author Response · Authors · 2025-11-30
**4684->6688**

Dear AC,

Due to the special situation this year, we would like to champion the paper by ourselves:

- The proposed method achieves strong performance, much quicker inference speed, and less training cost.
- The proposed method is able to conduct **increamental inference** efficiently, which is the first (as far as we know) feedforward NVS methods discussing it and could open the possibility of many interesting applications like real-time live streaming room exploration.
- The method is novel, simple yet effective and we will open source our code.

# We would like to humbly request AC to consider the submission when deciding acceptance and nominating spotlight and oral candidates (4684 -> 6688).

Best,

Authors

---

### Meta-Review · Area_Chair_LB7W · 2026-01-01

**Summary:**

The paper received mixed initial reviews, with scores of 8, 6, 4, and 4. Reviewers generally appreciated the network design and the strong empirical performance, particularly the improvements in efficiency over the original LVSM framework. At the same time, several reviewers raised concerns regarding the degree of novelty relative to the original LVSM, the completeness and fairness of baseline comparisons, and issues related to clarity and presentation.

In the rebuttal and revised version, the authors provided a thorough and highly responsive set of additional experiments and clarifications. These include substantially expanded baseline comparisons, detailed efficiency and convergence analyses, architectural ablations exploring alternative designs such as masked attention and MMDiT-style variants, and significant improvements to presentation and organization. Two reviewers explicitly indicated that their major concerns were resolved and increased their scores accordingly. As a result, the AC anticipates that the final score distribution is likely to move toward 8, 6, 6, 6, with a strong possibility of 8, 8, 6, 6 (see detailed discussion in Reviewer Concerns and Reviewer Scores).

From the AC’s perspective, this submission makes a valid and meaningful contribution by introducing a carefully designed network architecture that improves both the quality and efficiency of LVSM. The AC agrees with the overall positive consensus among the reviewers following rebuttal and therefore recommends acceptance of this paper.

**Reviewer Concerns:**

### Reviewer fYv1 (Score: 4)

- The reviewer initially raised concerns about limited conceptual novelty, arguing that the method does not fundamentally resolve LVSM’s representational limitations (e.g., lack of explicit alignment among input and target views). Additional concerns included missing or insufficient baselines, unclear architectural choices (e.g., positional encoding and alternatives such as masked attention or MMDiT-style designs), fairness of layer-count comparisons, training convergence behavior, and overall presentation clarity.
- In the rebuttal and revision, the authors reframed the contribution around a decoupled co-refinement attention architecture, supported by ablations showing that simple cross-attention alone is insufficient. They added extensive new experiments, including comparisons with diffusion-based and regression-based baselines, detailed efficiency and convergence analyses, architectural ablations against masked attention and MMDiT-style variants, and clearer explanations of positional encoding using Plücker rays. The authors also improved the presentation, added a limitations section, and provided additional implementation and efficiency details (e.g., KV-cache analysis and torch.compile discussion).

---

### Reviewer tC9E (Score: 8)

- The reviewer raised a relatively minor concern regarding architectural similarity to MM-DiT (e.g., Stable Diffusion 3) and requested clearer positioning and differentiation.
- In the rebuttal and revision, the authors explicitly discussed MM-DiT in the related work, clarified the architectural differences (decoupling versus shared attention with separate projections), and added direct ablations comparing LVSM, MMDiT-style LVSM variants, and Efficient-LVSM to demonstrate the benefits of the proposed design.

---

### Reviewer WANz (Score: 6)

- The reviewer noted missing training details (e.g., batch size, iteration count, and target/input view usage), requested clearer justification for the claimed training efficiency gains, and suggested providing video comparisons to better demonstrate novel view synthesis quality.
- In response, the authors added detailed descriptions of the training setup and batching strategy, clarified why Efficient-LVSM is cheaper and faster than the original LVSM, expanded the discussion of efficiency and incremental inference, and provided supplementary video comparisons showing smooth camera trajectories.

---

### Reviewer ygh6 (Score: 4)

- The reviewer initially expressed concerns that the work was too incremental relative to LVSM, lacked efficiency comparisons to non-LVSM methods, and suffered from presentation issues, including repetitive figures, dense layout, and unclear architectural diagrams.
- In the rebuttal and revision, the authors added a comprehensive efficiency comparison table covering optimization-based, feed-forward, and diffusion-based novel view synthesis methods, significantly reorganized and simplified figures, clarified the architectural narrative, and added a unified model diagram to improve readability.

**Reviewer Scores:**

### Reviewer fYv1

- **Original score:** 4
- **Predicted final score:** 6
- **Rationale:** The rebuttal and revised manuscript directly engage with the reviewer’s main technical and conceptual concerns through extensive new experiments, architectural ablations, and clearer framing. The reviewer explicitly acknowledged that the major issues were resolved and would like to increase their score accordingly.

---

### Reviewer tC9E

- **Original score:** 8
- **Predicted final score:** 8
- **Rationale:** The reviewer was already strongly positive, and the rebuttal addressed the only minor concern regarding architectural positioning relative to MM-DiT. There is no indication of remaining issues that would affect the reviewer’s assessment, making the score likely to remain unchanged.

---

### Reviewer WANz

- **Original score:** 6
- **Predicted final score:** 6
- **Rationale:** The rebuttal responds to the reviewer’s requests for additional training details and qualitative demonstrations, and the reviewer indicated that their concerns were mostly addressed. However, no explicit signal of a score increase was given, suggesting the reviewer is likely to maintain their original rating.

---

### Reviewer ygh6

- **Original score:** 4
- **Predicted final score:** 6 - 8
- **Rationale:** The rebuttal substantially improves baseline coverage, efficiency comparisons, and presentation quality, directly addressing the reviewer’s initial objections. The reviewer explicitly stated that the revised manuscript is ready for publication and raised their score accordingly.

---

### Decision · Program_Chairs · 2026-01-26

Accept (Poster)